# Vps60 initiates alternative ESCRT-III filaments

Anna-Katharina Pfitzner[1]*, Henry Zivkovic[1]*, César Bernat-Silvestre[1], Matt West[2], Tanner Peltier[2], Frédéric Humbert[1], Greg Odorizzi[2], and Aurélien Roux[1,3]

**Endosomal sorting complex required for transport-III (ESCRT-III) participates in essential cellular functions, from cell division to endosome maturation. The remarkable increase of its subunit diversity through evolution may have enabled the acquisition of novel functions. Here, we characterize a novel ESCRT-III copolymer initiated by Vps60. Membrane-bound Vps60 polymers recruit Vps2, Vps24, Did2, and Ist1, as previously shown for Snf7. Snf7- and Vps60-based filaments can coexist on membranes without interacting as their polymerization and recruitment of downstream subunits remain spatially and biochemically separated. In fibroblasts, Vps60/CHMP5 and Snf7/CHMP4 are both recruited during endosomal functions and cytokinesis, but their localization is segregated and their recruitment dynamics are different. Contrary to Snf7/CHMP4, Vps60/CHMP5 is not recruited during nuclear envelope reformation. Taken together, our results show that Vps60 and Snf7 form functionally distinct ESCRT-III polymers, supporting the notion that diversification of ESCRT-III subunits through evolution is linked to the acquisition of new cellular functions.**

## Introduction

Lipid membranes are a hallmark of living cells. To maintain functionality, they require constant remodeling by dedicated machineries, like the endosomal sorting complex required for transport-III (ESCRT-III). Presumed to be the first membrane remodeling machinery to have evolved (Spang et al., 2015; Liu et al., 2021), ESCRT-III acts on virtually all cellular membranes to promote membrane fission from within membrane necks, a process that is essential for many cellular functions, including the formation of intralumenal vesicles (ILVs) at endosomes, cytokinetic abscission of the plasma membrane, reformation of the nuclear envelope, and closure of autophagosomes (Schöneberg et al., 2017; Gatta and Carlton, 2019; Vietri et al., 2020). Moreover, ESCRT-III catalyzes budding of various virions in eukaryotes and archaea (Barnes and Wilson, 2019; Broniarczyk et al., 2017; Lippincott-Schwartz et al., 2017; Ortmann et al., 2008; Streck et al., 2018; Tabata et al., 2016) and functions in repairing lipid membranes, as shown for the eukaryotic and bacterial plasma, lysosomal, nuclear, and plastid membranes (Denais et al., 2016; Jimenez et al., 2014; Raab et al., 2016; Radulovic et al., 2018; Skowyra et al., 2018; Liu et al., 2021; Junglas et al., 2021; Gupta et al., 2021), a function essential to sustain vacuolar confinement of pathogens during certain infections (Göser et al., 2020; López-Jiménez et al., 2018). Unlike other membrane remodeling machineries, ESCRT-III can also function with a reverse orientation that promotes membrane fission from the outside of membrane necks during release of peroxisomes, recycling of endosomes, and lipid droplet formation (Allison et al., 2013; Chang et al., 2019; Mast et al., 2018). Despite its ubiquitous role in vital functions, the mechanism by which ESCRT-III performs membrane remodeling and, especially, the adaptation of the machinery to its various cellular functions is not fully understood.

Canonically recruited to endosomal membranes by ESCRT-II, ESCRT-III assembly starts with Vps20 (CHMP6) followed by subunits Snf7 (CHMP4B), Vps2 (CHMP2A), and Vps24 (CHMP3) before being likely completed by subunits Did2 (CHMP1B) and Ist1 (IST1). Besides the canonical pathway, other nucleators, such as Bro1 (ALIX) and Chm7 (CHMP7), can recruit ESCRT-III to diverse cellular membranes (Dores et al., 2012, 2016; Pashkova et al., 2013; Olmos et al., 2016; Vietri et al., 2015; Webster et al., 2016). Vps2 and Vps24 as well as Vps2 and Did2 bind Snf7 synergistically and then recruit the AAA-ATPase Vps4 (Babst et al., 2002; Teis et al., 2008, 2010; Saksena et al., 2009; Obita et al., 2007; Stuchell-Brereton et al., 2007; Brune et al., 2019), which induces subunit turnover within ESCRT-III polymers, promoting either disassembly (Lata et al., 2008; Adell et al., 2014; Yang et al., 2015), growth (Mierzwa et al., 2017), or

........................................................................................................................................................................
[1]Department of Biochemistry, University of Geneva, Geneva, Switzerland;   [2]Department of Molecular Cellular and Developmental Biology, University of Colorado, Boulder, CO, USA;   [3]National Center of Competence in Research in Chemical Biology, University of Geneva, Geneva, Switzerland.

*A.-K. Pfitzner and H. Zivkovic contributed equally to this paper.   Correspondence to Aurélien Roux: aurelien.roux@unige.ch;   Anna-Katharina Pfitzner: anna-katharina_pfitzner@hms.harvard.edu

A.-K. Pfitzner's current affiliation is Department of Cell Biology, Harvard Medical School, Boston, MA, USA.   H. Zivkovic's current affiliation is Max Planck Institute for Biochemistry, Martinsried, Germany.

sequential subunit polymerization (Pfitzner et al., 2020). In cells, Vps4-dependent polymer remodeling is indispensable for ESCRT-III function (Guizetti et al., 2011; Mierzwa et al., 2017; Adell et al., 2017). Upon recruitment, ESCRT-III subunits assemble into filaments with diverse stoichiometries and shapes, ranging from spirals (Henne et al., 2012; Shen et al., 2014; Chiaruttini et al., 2015; Hanson et al., 2008) to tubular helices (Lata et al., 2008; Effantin et al., 2013; Nguyen et al., 2020; McCullough et al., 2015; Hanson et al., 2008) and spiraling membrane tubes (Moser von Filseck et al., 2020; Bertin et al., 2020). Sequential succession of these various ESCRT-III filaments has recently been suggested to promote ESCRT-III–mediated membrane remodeling (Harker-Kirschneck et al., 2022; Pfitzner et al., 2021; Jiang et al., 2022).

Aside from the well-characterized core subunits Snf7, Vps2, and Vps24, several accessory ESCRT-III subunits have been identified based on a deletion phenotype indicative of disturbed ILVs formation (Katzmann et al., 2001) and their secondary structure organization (Leung et al., 2008), which is highly conserved among ESCRT-III proteins even across species. As one of those accessory subunits, the function of Vps60/Mos10 (CHMP5), though briefly associated with ESCRT-III disassembly (Nickerson et al., 2010; Azmi et al., 2008), remains poorly understood to this day. A recent analysis of genetic interactions between ESCRT-III subunits, however, places Vps60 more centrally in an interaction network (Brune et al., 2019), implying a more important function for Vps60 than previously recognized. We thus decided to perform a functional characterization of Vps60 and its interactions with other ESCRT-III subunits as well as the ATPase Vps4.

## Results

### Vps60 behaves like an early ESCRT-III protein

We here set out to characterize the function of ESCRT-III accessory subunit Vps60. In general, most ESCRT-III subunits or submodules can, to varying degrees, polymerize into membrane-bound filaments which often depict preferential binding to a specific membrane curvature range (Chiaruttini et al., 2015; De Franceschi et al., 2018; Bertin et al., 2020). To analyze Vps60's membrane-binding properties, we, thus, injected labeled ESCRT-III subunits in the vicinity of membrane nanotubes, the latter made by pulling beads adhered to giant unilamellar vesicles (GUVs) with optical tweezers (see Materials and methods and Fig. 1 A and Fig. S1, B–E). This assay produces highly curved and flat membranes close to each other, allowing us to evaluate curvature-dependent binding in a wide range. As previously reported, Snf7, the initial ESCRT-III subunit, bound exclusively to flat membrane (Fig. 1 A and Fig. S1 B; De Franceschi et al., 2018; Chiaruttini et al., 2015). Likewise, Vps60 is strongly recruited along the GUV's flat membrane, whereas binding is barely observable along the highly curved nanotube (Fig. 1 A and Fig. S1 C). In contrast, both downstream ESCRT-III submodules, Vps2-Vps24 and Vps2-Did2-Ist1, bind predominantly to highly curved nanotubes (Fig. 1 A; Fig. S1, D and E). Vps60 puncta observed outside the nanotube or GUV outlines most likely correspond to protein polymers as they do not contain membrane (Fig. S1 C). Interestingly, Alexa488-

Vps60, analogous to Alexa488-Snf7, polymerized spontaneously (Fig. S1 G) on GUVs and was recruited efficiently to flat non-deformable supported lipid bilayers (SLBs; Fig. 1 B), whereas downstream submodules Vps2-Vps24 (Teis et al., 2008; De Franceschi et al., 2018) and Vps2-Did2-Ist1 (Pfitzner et al., 2020) required activation, here provided by an acidic buffer (Fig. S1 G), to polymerize along nanotubes, and membrane binding was absent on SLBs (Fig. S1 F).

Upon membrane binding, all characterized ESCRT-III subunits polymerize into filaments (Henne et al., 2012; Shen et al., 2014; Chiaruttini et al., 2015; Hanson et al., 2008; Lata et al., 2008; Effantin et al., 2013; Nguyen et al., 2020; Moser von Filseck et al., 2020). To test if Vps60 behaves likewise, we next performed negative-stain EM of large unilamellar vesicles (LUVs) incubated with Vps60. Indeed, Vps60 formed ring-shaped filaments with an average diameter of 30.2 ± 3.4 nm (Fig. 1, C and D) which bound to membrane via an interface perpendicular to their radial axis (Fig. 1 E), similar to ring filaments described for Snf7 (Henne et al., 2012; Chiaruttini et al., 2015). In disagreement with our findings, a recent study reported Vps60 to form wide-ranging spirals reminiscent of Snf7 spirals (Banjade et al., 2021). This discrepancy might arise from different experimental conditions as Banjade and colleagues used higher protein concentrations and different liposome compositions, which might help propagate spiral growth. Interestingly, some polymers possessed one inward-curled tip, as would be expected for a spiral initiator (Fig. 1 C). Snf7 spiral polymers were previously suggested to grow out of ring-shaped filaments upon their spontaneous breakage (Chiaruttini et al., 2015; Lenz et al., 2009). In analogy, curled Vps60 polymers might arise from rupture of ring filaments. The lack of large spirals, however, indicates Vps60-specific filament's properties preventing further polymerization into spirals. Alternatively, curled filaments might result from ring breakage during sample preparation. Besides rings, we also observed some rode-like structures (Fig. 1 F), which however were never associated with membrane. We speculate that this might be due to spontaneous polymerization of Vps60 either in absence of membrane or due to detachment of the filaments from the membrane after initial nucleation at its surface.

In summary, Vps60 depicts characteristics similar to the early ESCRT-III protein Snf7 and clearly distinct from the properties of downstream modules Vps2-Vps24 and Vps2-Did2-Ist1, sharing its curvature preference (flat), similar filament shape (mostly ring/spiral), and a membrane-binding interface perpendicular to the helical axis (Moser von Filseck et al., 2020; Tang et al., 2016; Fig. 1 G). The higher-curvature binding preference of later subunits (Vps2-Vps24, Did2-Ist1), in contrast, cohere with the helical polymers formed and a membrane-binding interface parallel to the helical axis (Nguyen et al., 2020; McCullough et al., 2015; Lata et al., 2008; Effantin et al., 2013; Azad et al., 2023).

### Spontaneous nucleation of Vps60 on membrane is highly efficient

As both Vps60 and Snf7 polymerize spontaneously on membrane, we next set out to compare their nucleation capacities.

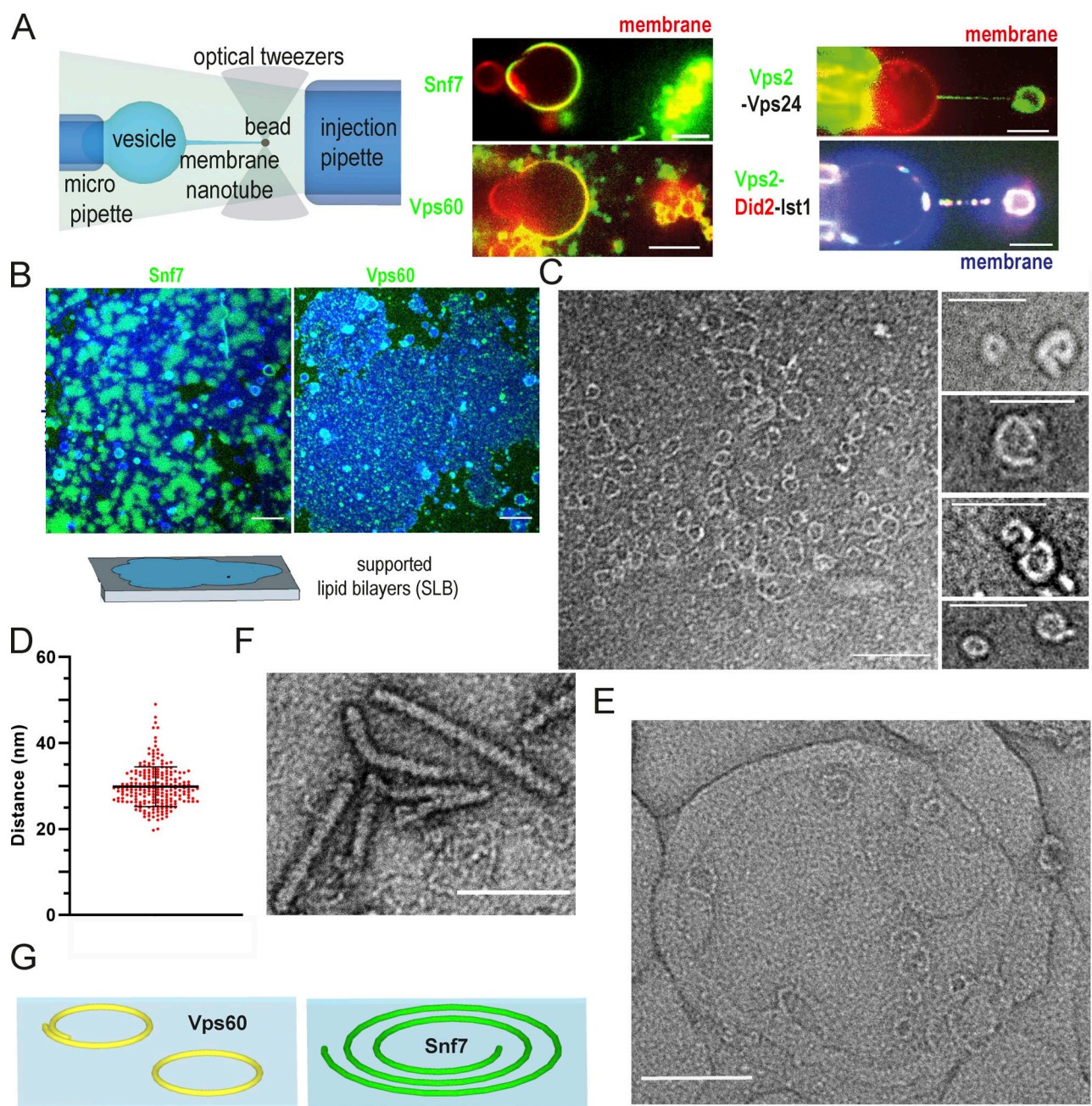

Figure 1. **Comparison of biochemical properties of ESCRT-III proteins. (A)** Schematic representation of membrane nanotube pulling and confocal microscopy images of Snf7-Alex488 (green), Vps60 (green), Vps2-Alex488 (green), and Vps24 or Vps2-Alex488 (green) and Did2-Atto565 (red), and Ist1 binding to membrane nanotubes (red, blue; scale bar: 5 μm). **(B)** Confocal images of Snf7 (green) or Vps60 (green) nucleation on SLBs (red; scale bar: 10 μm). **(C)** Negative stain electron micrographs of Vps60 filaments polymerized on LUVs (scale bar: 100 nm). **(D)** Quantification of the experiment described in C (n = 7, ROI = 69). **(E)** Negative stain electron micrographs of Vps60 filaments polymerized on LUVs (scale bar: 100 nm). **(F)** Negative stain electron micrographs of Vps60 filaments (scale bar: 100 nm). **(G)** Schematic representation of Vps60 filaments.

We, therefore, analyzed the nucleation rate of Atto565-Vps60 on SLBs (Fig. 2, A and B). Below 50 nM, Atto565-Vps60 nucleation events increased linearly with respect to protein concentration, whereas above 50 nM, their number increased exponentially. In contrast to the estimated nucleation rate, which was likely underestimated due to overlapping Vps60 puncta, images at concentrations between 150 and 250 nM do not suggest a saturation of Vps60 binding. These results point toward Vps60 displaying a higher intrinsic nucleation rate on membrane than Snf7, the latter of which does not spontaneously nucleate below a concentration of 300 nM. Vps60 does not form growing patches like Snf7, but instead, its binding manifests itself in accumulation of puncta (Fig. 2 C) as well as an overall increase of intensity on the membrane. Growth of Snf7 patches was previously explained by breaking preexisting spirals into multiple smaller spirals from which protein polymerization could continue (Chiaruttini et al.,

Figure 2. **Nucleation and polymerization properties of Vps60 on membrane. (A)** Confocal images of SLBs incubated with indicated concentration of Vps60 (scale bar: 10 μm). **(B and C)** Quantification of Vps60 puncta per area at different Vps60 concentrations from the experiment described in A. ROI ≥ 50.

**(C)** Time-lapse experiments of SLBs (gray) incubated with Vps60 (red, upper panel) or Snf7 (green, lower panel). Scale bar: 10 µm. **(D)** Confocal images of SLBs incubated with Vps60 (black) and Snf7 (green) were indicated (scale bar: 10 µm). **(E)** Quantification of experiments described in D (n = 3, ROI ≥ 6). **(F)** Confocal images of SLBs (blue) incubated with Snf7 (red) and Vps60 (green; scale bar: 2 µm; n = 3). **(G)** Plot of fluorescence profile of a representative membrane section from experiments described in F. **(H)** Confocal images of time-lapse experiment of addition of Vps60 (red) to SLBs (blue) preincubated with Snf7 (green), Vps2, Vps24, Did2, and Ist1 (n = 3). Scale bar: 10 µm. **(I)** Plot of fluorescence profile of an exemplary membrane section (t = 20 min) from experiments described in H.

2015). Filament breaking thus fuels a chain reaction from a single nucleation event, leading to expanding protein patches formed by hundreds of growing spirals. Observed filament structures of Vps60 (Fig. 1 C) suggested that filament breaking might occur less frequently and that spiral growth was partially or completely inhibited, explaining why no growing Vps60 patches were observed (Fig. 2 C).

### Vps60 and Snf7 display mutually exclusive membrane binding patterns

To next address if Vps60 and Snf7 interact upon each other's polymerization on membrane, we analyzed Vps60 nucleation on SLBs preincubated with Snf7, simultaneous polymerization of both proteins and in absence of Snf7, overall revealing that Vps60 binding remains unaffected by the presence of Snf7 (Fig. 2, D and E; Fig. S2, A and B). Similarly, Snf7 patches grew normally on SLBs preincubated with Vps60 (Fig. S2 C). In fact, no colocalization of Snf7 and Vps60 was observed, even when the whole membrane surface was covered, indicating that Vps60 and Snf7 membrane binding are mutually exclusive (Fig. 2, F and G; Fig. S2 D). Moreover, incubation of Snf7 patches with high concentrations of Vps60 resulted in a decrease in Snf7 intensity on the membrane and vice versa (Fig. S2, E–H), which suggests Vps60 and Snf7 compete to bind on available membrane surface.

Following polymerization, Snf7 filaments recruit downstream subunits starting with Vps2-Vps24 (Mierzwa et al., 2017), followed by Vps2-Did2, and finally Ist1 (Pfitzner et al., 2020). As Vps60 and Snf7 polymers coexist on membrane without observable interaction, we asked if Vp60 is recruited into Snf7-based polymers by downstream subunits. We saw no recruitment of Vps60 to Snf7 polymers in the presence of Vps2, Vps24, Did2, and Ist1 or when Snf7 patches were preincubated with all the downstream ESCRT-III subunits (Fig. 2, H and I; Fig. S3, A and B). Instead, Vps60 bound membrane identically in the absence or presence of any downstream subunits. Likewise, no integration of Vps60 into Snf7 polymers was seen upon Vps4-induced filament turnover, even in the presence of downstream subunits (Fig. S3, C and D).

In summary, we find Vps60 polymers and Snf7 polymers coexist independently on membranes, with no recruitment of Vps60 to Snf7-based polymers. Altogether, with its characteristics similar to Snf7, we wondered if Vps60 might function in parallel with Snf7 as an alternative initiating subunit for a multisubunit ESCRT-III filament.

### Vps60 polymers recruit downstream ESCRT-III subunits

To test our hypothesis, we studied ESCRT-III subunit binding to SLBs preincubated with Vps60. Indeed, Alexa488-Vps2 was recruited strongly to Atto565-Vps60–covered SLBs in the presence of Vps24 and Did2 (Fig. 3, A and B). Similarly, we observed Vps60-

mediated membrane binding of Alexa488-Vps24 in the presence of Did2 and Vps2 (Fig. 3, C and D) and recruitment of Alexa488-Did2 when supplemented with Vps2 and Vps24 (Fig. 3 E and Fig. S4 A). While Vps2-Vps24 are recruited to Snf7 polymers, Vps2-Vps24 was not sufficient for binding to Vps60-covered SLBs, though, we observed mild recruitment of Vps2-Did2 (Fig. 3, A and E). Vps24, however greatly increased Vps2-Did2 binding efficiency to Vps60-covered membranes, indicating that Vps24 might promote Vps2-Did2 heterofilament formation. As Vps60 and Did2 are reported to interact with Vps4-cofactor Vta1, we tested if Vta1 affects the recruitment of Vps2-Did2-Vps24 to Vps60, which was not the case (Fig. S4, B and C). Overall, we do not observe specific binding of Vps2, Vps24, and Did2 to Vps60 puncta, but find that their binding to the membrane is specifically enhanced in the presence of Vps60 puncta. As Vps60 filaments do not form patches, it may be that Vps60 serves as a nucleation template for ESCRT-III polymers which can then diffuse along membranes.

Alexa488-Ist1, analogous to the other ESCRT-III subunits, was specifically recruited to Vps60-covered vesicles in the presence of Vps2, Vps24, and Did2 (Fig. 3 F and Fig. S4 D). For this experiment, we used GUVs instead of SLBs as Ist1 formed aggregates in solution which sedimented on the SLBs, precluding the monitoring of Vps60-induced binding. The Ist1 binding pattern mirrored Did2 recruitment, which is consistent with the previously reported Did2-Ist1 heterodimer formation (Rue et al., 2008) and suggests Ist1 incorporation into ESCRT-III polymers relies on its interaction with Did2.

During Snf7-mediated recruitment of downstream subunits, distinct recruitment kinetics can be observed. We thus asked if a similar temporal organization can be seen for Vps60-induced membrane binding. Time-lapse imaging of Vps60-covered SLBs incubated with Vps2, Vps24, Did2, and Ist1 revealed a synchronic increase in Vps2, Vps24, and Did2 intensity and a slightly delayed increase in Ist1 intensity (Fig. 3 G). Taken together, this result suggests Vps2, Did2, and Vps24 together form the initial recruitment complex (Fig. 3 H). Subsequently, membrane-bound Did2 then triggers binding of Ist1 to ESCRT-III polymers. Compared with Snf7-mediated recruitment of Vps2-Vps24, binding of downstream subunits is slower when initiated by Vps60. Indeed, recruitment kinetics of Vps2-Vps24-Did2 to Vps60 filaments appears equivalent to Vps2-Did2 binding to Snf7 polymers (Pfitzner et al., 2020), supporting the notion that in Vps60-mediated assemblies, the two initial waves of subunits (Vps2-Vps24, and then Vps2-Did2) are condensed into a single Vps2-Vps24-Did2 wave.

### Vps60 nucleates distinct ESCRT-III copolymers

Snf7 recruitment of downstream ESCRT-III subunit results in formation of membrane-associated copolymers (Mierzwa et al., 2017; Moser von Filseck et al., 2020). As Vps60, likewise, was

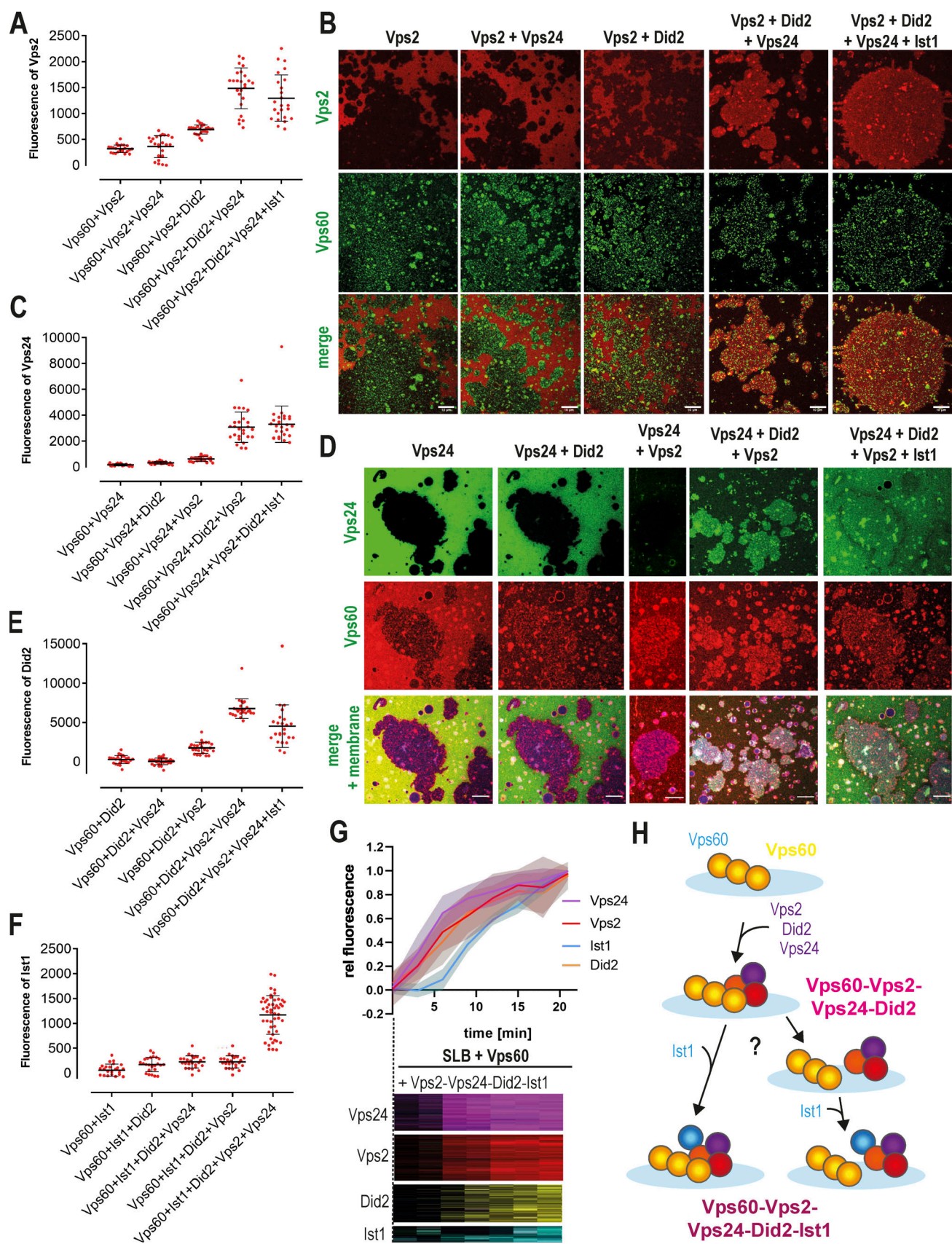

Figure 3. **Vps60 recruits ESCRT-III subunits to membrane. (A and B)** Quantification of Vps2 fluorescence (A, $n \geq 3$, ROI $\geq 23$) and confocal images (B) of Vps60-covered SLBs incubated with Alexa488-Vps2 and the indicated protein mixture. **(C and D)** Quantification of Vps24 fluorescence (C, $n = 3$, ROI $\geq 24$) and

confocal images (D) of Vps60-covered SLBs incubated with Alexa488-Vps24 and the indicated protein mixture. **(E)** Quantification of Did2 fluorescence of Vps60-covered SLBs incubated with Alexa488-Vps24 and the indicated protein mixture ($n$ = 3, ROI ≥ 24). **(F)** Quantification of Ist1 fluorescence of Vps60-covered GUVs incubated with Alexa488-Vps24 and the indicated protein mixture ($n$ = 3, ROI ≥ 24). **(G)** Kymographs and quantification of time-lapse experiments of Vps60-covered SLBs incubated with Alexa488-Vps24 (purple), Alexa488-Vps2 (red), Alexa488-Did2 (yellow), and Alexa488-Ist1 (cyan; $n$ ≥ 3, ROI ≥ 24). **(H)** Schematic representation of the model for sequential binding of ESCRT-III subunits to Vps60-polymers ($n$ ≥ 3, ROI ≥ 29). Scale bar: 10 μm.

able to recruit Vps2, Vps24, and Did2 to SLBs, we wondered whether similar cofilaments could be nucleated by Vps60. We thus performed negative stain electron microscopy of LUVs incubated with Vps60 and diverse subsets of ESCRT-III subunits (Fig. 4, A–C, and Fig. S3, E–G). Intriguingly, whereasVps60 alone formed mostly single-stranded rings (Fig. 4, A and D), co-incubation of LUVs with Vps60, Vps2, and Vps24 resulted in double-stranded spirals (Fig. 4, B and D; Fig. S3 E), and three-stranded spirals and rings were observed in presence of Vps60-Vps2-Vps24-Did2 (Fig. 4, C and D; Fig. S3 F). The linearly increasing filament thickness (Fig. 4 D) from 4.3 ± 0.5 nm (Vps60) to 8.4 ± 0.7 nm (Vps60-Vps2-Vps24) and 12.5 ± 1.3 nm (Vps60-Vps2-Vps24-Did2) strongly indicates the addition of a lateral strand analogous to observations in Snf7-based polymers (Mierzwa et al., 2017). The transformation of rings into spirals suggests that Vps60 rings break open upon binding of subsequent subunits. More interestingly, in samples including Vps60, Vps2, and Vps24, we also observed a two-stranded double helical polymer with a ring at one end (Fig. 4 E and Fig. S3 G). Occasionally, branching of these polymers was observed. We speculate that these double-helical polymers, which present similar to Vps2-Vps24–only polymers, might arise from nucleation and outgrowth of an initial Vps60 ring in cases when the polymer detached from its membrane support prior to or during the polymerization (Henne et al., 2012; Moser von Filseck et al., 2020). In support of this notion, double helical polymers, in contrast to spiral polymers, were never found associated with membranes. Furthermore, that a double-stranded ring is connected to a helical polymer likely suggests a non-homogeneous distribution of subunits within the Vps60-based ESCRT-III copolymer, e.g., Vps60 forming the ring structure while Vps2-Vps24 is dominating the double helical structure. Such a potential inhomogeneous subunit distribution might account for the non-perfect colocalization of Vps60 and downstream subunits observed in our fluorescence-based SLB-recruitment assay. This may also explain why we did not observe detachment of ESCRT-III filament from an initial Vps60 nucleator polymers in our EM data. Another difference between our EM data and our fluorescence membrane recruitment data, which suggest that Vps2, Vp24 as well as Did2 are required for efficient recruitment to Vps60-based polymer, is that we observed the formation Vps60-Vps2-Vps24 copolymers in our EM experiments. This might due be to longer incubation times and higher protein concentrations used in the EM experiment. Furthermore, we also find large numbers of membraneless double helical polymers in Vps60-Vps2-Vp24 samples not present inVps60-Vps2-Vp24-Did2 samples. This could imply a proneness of Vps60-Vps2-Vps24 filaments to form in solution or detach easily from membranes. These events might be missed in the SLBs assay, which only observes large filaments that stay attached to the membrane.

Overall, our results show that membrane-bound Vps60 ring filaments are able to form copolymers with downstream ESCRT-III subunit Vps2, Vps24, and Did2, transforming into multi-stranded spiral polymers during this process. Intriguingly, the heterogenous filament structure of double helical Vps60-Vps2-Vps24 polymers might hint toward inhomogenous subunit distribution within ESCRT-III polymers, potentially offering new insight into ESCRT-III filament organization and their molecular functioning during membrane remodeling activity.

### Vps60-nucleated ESCRT-III polymers undergo Vps4-mediated turnover

Previous studies demonstrated that ESCRT-III function in cells crucially depends on the ATPase activity of Vps4, which triggers filament turnover or remodeling (Adell et al., 2017; Mierzwa et al., 2017; Guizetti et al., 2011). To study if Vps60-based polymers are likewise remodeled by Vps4, we performed time-lapse imaging of SLBs preincubated with Vps60, Vps2, Vps24, and Did2. Upon addition of Ist1, Vps4, and ATP, Vps2 and Vps24 intensities decreased rapidly, indicating subunit disassembly (Fig. 5 A). In contrast, Did2 and Ist1 remained stably bound to membranes when Ist1 was in excess, whereas Vps4-triggered disassembly occurs in the absence of Ist1 (Fig. 5 A; Fig. S5, A and B). These results suggest that Did2 is protected from disassembly by competitive binding of Ist1 and Vps4 to Did2 (Fig. S5 E), a mechanism that we previously proposed for Snf7-based filaments (Pfitzner et al., 2020). As bound Ist1 itself undergoes continuous Vps4-triggered turnover, equilibrium between free Ist1 and Vps4 probably determines if Did2 is disassembled or stabilized.

Importantly, Vps60, unlike Snf7, remained bound to the membrane upon incubation with Vps4/ATP or Ist1-Vps4/ATP (Fig. 5 B). Moreover, neither supplementation with Vps4 cofactor Vta1 nor Bro1, which were both shown to interact with Vps60 (Shen et al., 2016; Yang et al., 2012), resulted in a disassembly of Vp60 from the membrane (Fig. 5 B). Likewise, Vps60-polymers remained stable with a 10-fold increased Vps4 concentration, which may overcome a lower sensitivity of Vps60 filaments toward Vps4. No direct disassembly of Vps60 by Vps4 in the absence of any downstream ESCRT-III subunits was observed (data not shown).

While Vta1 did not promote Vps4-mediated Vps60 disassembly, it did affect the depolymerization of downstream subunits from Vps60-based polymers. Indeed, Vta1 shifted Ist1's equilibrium from binding to disassembly (Fig. S5, D and E), therefore perturbing Ist1-mediated protection of Did2 from Vps4-mediated disassembly (Fig. S5, C and E). Additionally, Vps2 and Vps24 depolymerization rates increased (Fig. 5, C and D). Intriguingly, in the presence of Vta1, Vps2 and Vps24 depolymerization (Fig. 5, C and D) increased much more than did the disassembly of Did2 (Fig. 5 E and Fig. S5 C). Overall, these

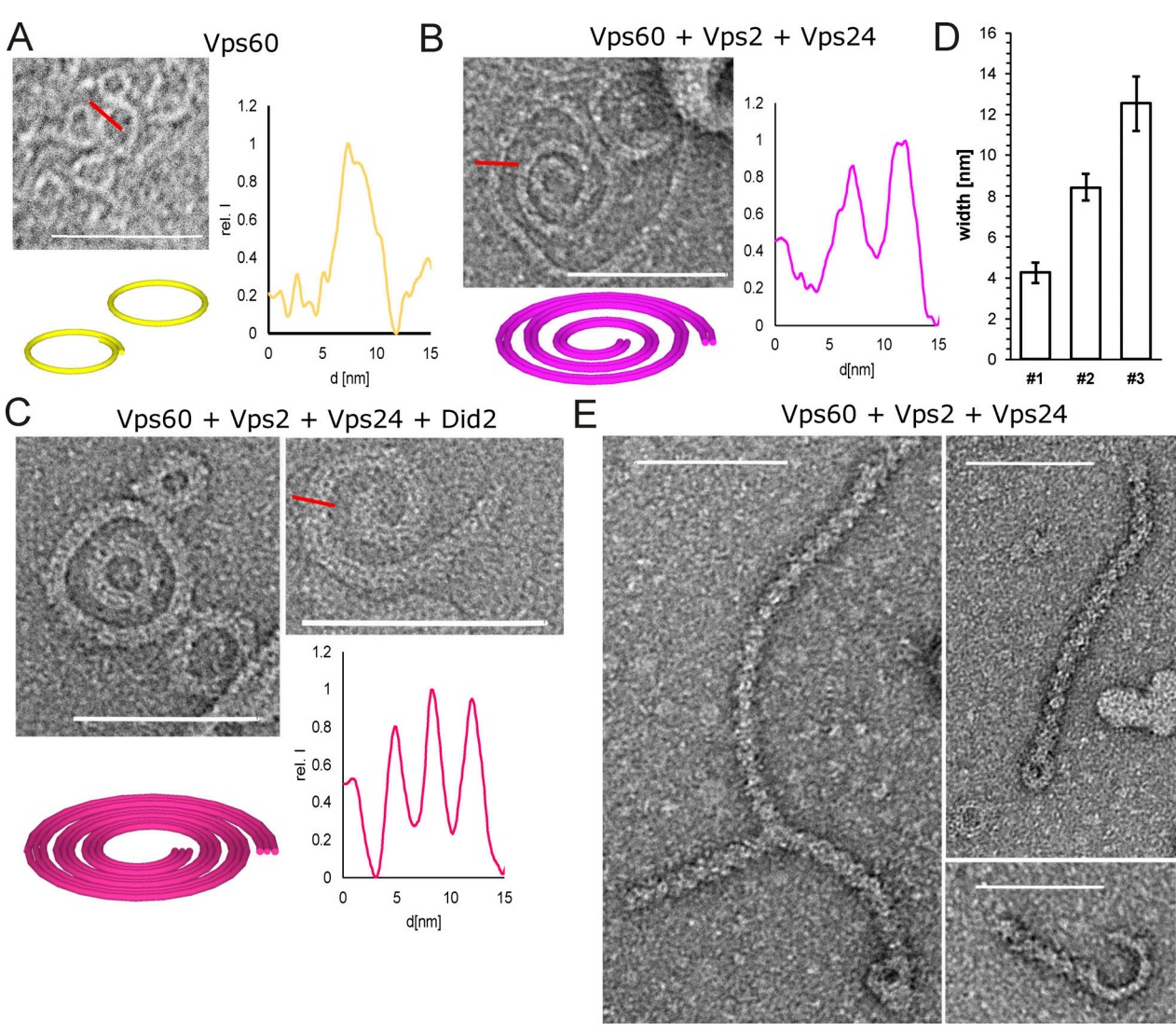

Figure 4. **Vps60 triggers ESCRT-III copolymer formation. (A–C)** Negative stain electron micrographs of Vps60 filaments incubated with indicated proteins polymerized on LUVs (scale bar: 100 nm), representative profile scan of filaments, and schematic representation of observed polymers. **(D)** Quantification of filament width from experiments described in A–C (n = 3, ROI = 285 [Vps60]; ROI = 56 [Vps60–Vps2–Vps24]; ROI = 21 [Vps60–Vps2–Vp24–Did2]). **(E)** Negative stain electron micrographs of Vps60-Vps2-Vps24. Scale bar: 100 μm.

results suggest Vta1 potentially targets specific subunits to increase disassembly. Alternatively, Vta1-binding may primarily increase Vps4 activity, while differences between subunit disassembly rates may rely more on their accessibility to Vps4 within the polymer structure. While all three subunits display simultaneous and synchronic membrane recruitment, Vta1's strong influence on Vps2 and Vps24 depolymerization rates compared with Did2 establishes a depolymerization hierarchy between these three subunits. It is tempting to speculate that such Vta1-induced divergence of disassembly rates could promote the assembly of ESCRT-III subunits in a temporal sequence (Fig. 5 G) as we have previously shown for Snf7-based polymers (Pfitzner et al., 2020).

**Vps60- and Snf7-based polymers exert varying and distinct dynamic properties**

Vps60, like Snf7, can recruit downstream subunits to form a heteropolymer. To test whether Snf7 and Vps60 polymers compete during subunit recruitment, we incubated Alexa488-Snf7– and Atto565-Vps60–covered SLBs with Atto647-Vps2, Vps24, and Did2 (Fig. 5 H and Fig. S5 F). As a control, we incubated analogous SLBs with Vps2-Vps24 since it binds only to Snf7 patches and not to Vps60-covered membranes under condition used in our SLB assay (Fig. 5 I and Fig. S5 G). Upon recruitment of Vps2-Did2-Vps24, Vps2 fluorescence initially colocalized with Snf7 patches, followed by a slower binding to the Vps60-covered membrane (Fig. 5, H, J, and K). In contrast, Vps2-Vps24 only colocalized with Snf7 patches, as no recruitment to the Vps60-covered membrane was observed (Fig. 5, I–K). The two-stepped binding of Vps2-Did2-Vps24 likely emerges from an initial recruitment of Vps2-Vps24 to Snf7 patches. Thereafter, Vps2-Did2-Vp24 bind to Vps60 filaments by an independent recruitment process following slower kinetics (Fig. 3 G), implying that recruitment to Snf7- or Vps60-based filaments occurs independently of each other. To similarly

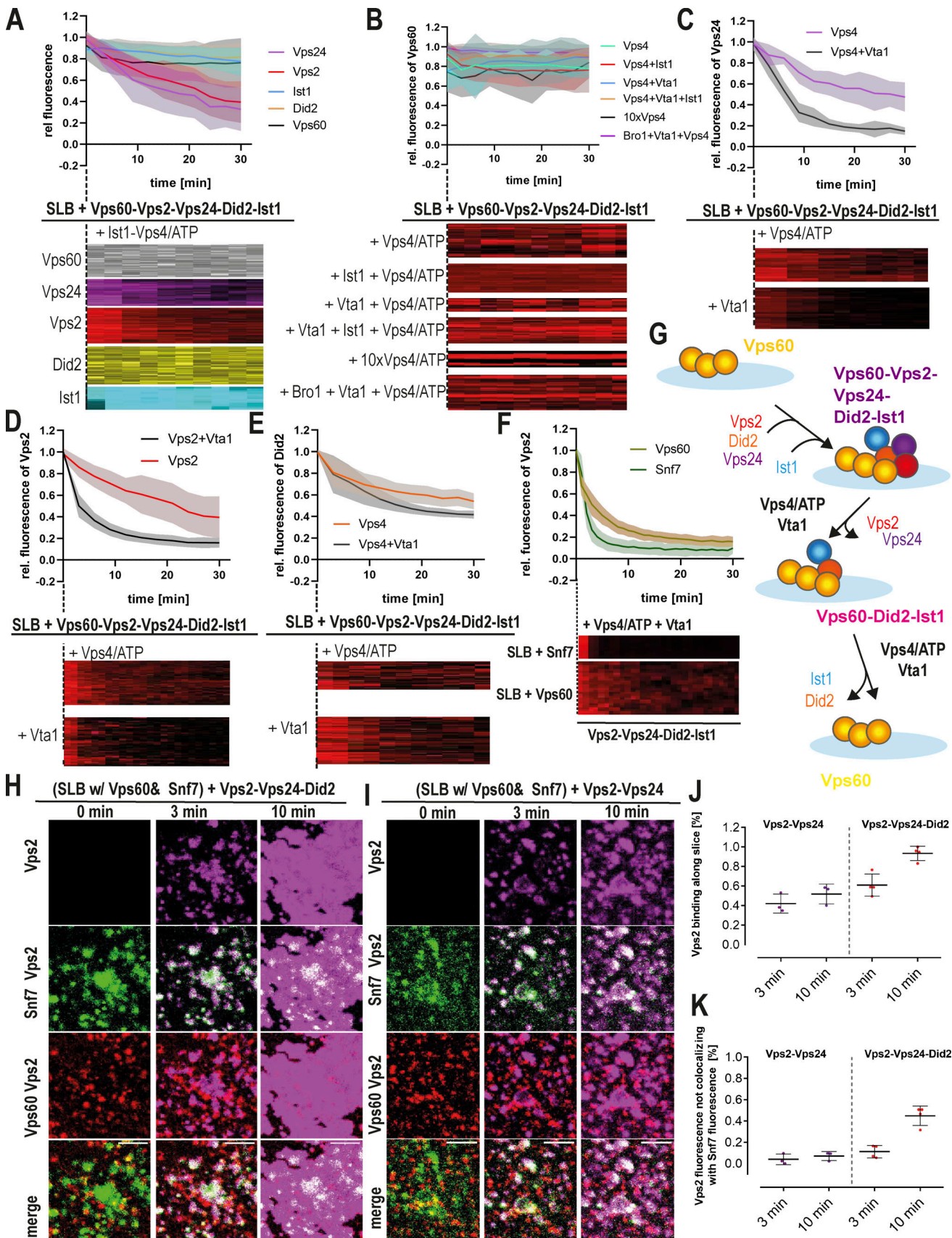

Figure 5. **Vps4 triggers turnover of Vps60-based ESCRT-III polymers. (A)** Quantification of fluorescence intensities of indicated subunit and kymographs of time-lapse experiments of addition of Ist1 and Vps4/ATP to SLBs preincubated with Vps60, Vps2, Vps24, Did2, and Ist1 ($n \geq 3$, ROI $\geq 30$). **(B)** Quantification of

fluorescence intensities of Vps60 and kymographs of time-lapse experiments of the addition of Vps4/ATP and the indicated proteins to SLBs preincubated with Vps60, Vps2, Vps24, Did2, and Ist1 (*n* ≥ 3, ROI ≥ 20). **(C)** Quantification of Vps24 fluorescence intensities and kymographs of time-lapse experiments of the addition of Vps4/ATP and Vta1 (when indicated) to SLBs preincubated with Vps60, Vps2, Alexa488-Vps24, Did2, and Ist1 (*n* = 3, ROI ≥ 18). **(D)** Quantification of Vps2 fluorescence intensities and kymographs of time-lapse experiments of the addition of Vps4/ATP and Vta1 (when indicated) to SLBs preincubated with Vps60, Vps24, Alexa488-Vps2, Did2, and Ist1 (*n* = 3, ROI ≥ 35). **(E)** Quantification of Did2 fluorescence intensities and kymographs of time-lapse experiments of the addition of Vps4/ATP and Vta1 (when indicated) to SLBs preincubated with Vps60, Vps2, Vps24, Alexa488-Did2, and Ist1 (*n* = 3, ROI ≥ 40). Control experiment without Ist1 together with Vps4 is shown in Fig. S5 A with regard to the data shown here. **(F)** Quantification of Vps2 intensity and kymographs of time-lapse experiments of addition of Vta1 and Vps4/ATP to SLBs preincubated with Vps2, Vps24, Did1 Ist1, and Snf7 or Vps60 (*n* = 3, ROI ≥ 80). **(G)** Cartoon of the model for Vps4-triggered sequential disassembly of Vps60-based ESCRT-III polymer. **(H)** Confocal images of time-lapse experiments of the addition of Atto647-Vps2, Vps24, and Did2 to SLBs preincubated with Snf7 (green) and Vps60 (red). Single channels are shown in Fig. S5 F. **(I)** Confocal images of time-lapse experiments of the addition of Atto647-Vps2 and Vps24 to SLBs preincubated with Snf7 (green) and Vps60 (red). Single channels are shown in Fig. S5 G. **(J and K)** Quantification of Vps2 fluorescence intensities total membrane coverage and non-colocalization with Snf7 patches from experiments described in H and I (*n* ≥ 3). Scale bar: 10 μm.

---

compare disassembly from Snf7- and Vps60-based ESCRT-III polymers, we monitored fluorescence of Vps2 upon addition of Vta1 and Vps4/ATP to SLBs preincubated with labeled Vps2, Vps24, Did2, and Vps60 or Snf7, respectively (Fig. 5 F). Vps2 depolymerization from Vps60-based polymers was slightly delayed compared with disassembly from Snf7-based filaments.

In conclusion, Vps60- and Snf7-based polymers assemble and undergo disassembly independently of each other. These results might indicate that both polymers can coexist in cells in separate functions. In direct comparison, Vps60-based filaments display delayed assembly and disassembly kinetics, which might indicate adaptation to cellular functions in which slower ESCRT-III assembly is required. Alternatively, we might miss cofactors in our in vitro reconstitution approach which could speed up the assembly and disassembly of Vps60-based polymers.

### Vps60 deletion reduces ILV formation in vivo

As our findings suggest that Vps60 and Snf7 form independent ESCRT-III polymers, we next wondered how Vps60 loss of function would affect ESCRT-III activity in cells during one of the most studied ESCRT-III functions, the formation of ILVs at endosomes. To address this, we performed cryo-fixation and dual-axis EM tomography of wildtype *Saccharomyces cerevisiae* or mutant yeast cells in which the *VPS60* gene had been deleted (*vps60Δ*). In general, the overall architecture of endosomes is not altered in *vps60Δ* cells (Fig. 6, A–D; and Videos 1 and 2). However, examination of individual endosomes revealed that the number of ILVs per endosome is reduced greater than fivefold in *vps60Δ* cells compared with wildtype cells (Fig. 6 E). This observation indicates that ILV formation remains partially functional in absence of Vps60, which is consistent with multiple pathways for ILV budding and scission driven by diverse ESCRT-III machineries.

To further investigate the functioning of Vps60 in vivo, we next tested its polymerization status by rate-zonal centrifugation of detergent-solubilized yeast membranes. This assay indicates the abundance and size of ESCRT-III polymers at steady state (Teis et al., 2008). As previously reported (Teis et al., 2008; Johnson et al., 2017), membrane-associated Snf7 in wildtype yeast is found predominantly in ~440-kD polymers, whereas the disruption of Vps4 ATPase activity, either by deletion of the *VPS4* gene or by replacement of *VPS4* with the catalytically inactive *vps4*[E233Q] allele, resulted in a shift of Snf7 toward higher molecular-weight polymers (Fig. 6, F and G). Formation of high-

molecular-weight Snf7 polymers in *vps4*-mutant cells occurs due to lack of Vps4-mediated disassembly and dissociation of Snf7 from membranes (Babst et al., 2002; Teis et al., 2008). Like Snf7, HA-tagged Vps60 was predominantly observed at 440 kD in wildtype yeast (Fig. 6 F), suggesting that Vps60 forms polymers similar in size and constitution to Snf7 filaments. Unlike Snf7, however, Vps4 inactivation shifted the distribution of membrane-associated Vps60-HA toward the unpolymerized state observed in lower-molecular-weight fractions of the density gradient (Fig. 6, F and G). Lack of accumulation of high-molecular-weight Vps60 filaments upon blocking Vps4 activity aligns with our in vitro observation that polymerized Vps60 is not turned over by Vps4 (Fig. 5 B) and might be suggestive of a divergent method of Vps60 recycling in cells.

Overall, these results suggest Vps60 occurs as polymers in vivo that operate differently from Snf7 polymers.

### CHMP5 displays a distinct recruitment pattern from CHMP4B

To further understand the role of Vps60 during ESCRT-III activity in vivo, we next decided to assay the recruitment pattern and dynamics of its mammalian homolog CHMP5 as ESCRT-III functions in mammalian cells are more diverse. Similar to other ESCRT-III subunits (Mercier et al., 2020; Skowyra et al., 2018; Radulovic et al., 2018), CHMP5 is recruited to endosomes upon osmotic shock (Fig. 7 A) and to lysosomes upon induced lysosomal damage via L-leucyl-L-leucine methyl ester (LLOMe) treatment (Fig. 7, B and C). The number and intensity of CHMP5 puncta per cell increased up to 30 min after osmotic shock treatment followed by a shallow decline (Fig. 7, D and F), which represents a clear delay compared with the 15-min peak and rapid decline in puncta fluorescence observed for CHMP4B (Mercier et al., 2020). Likewise, slower recruitment dynamics were observed for cells treated with LLOMe (Fig. 7, E and F). Moreover, similar to our in vitro data, CHMP5 puncta persistence during osmotic shock treatment exceeded that of CHMP4B though a decline was eventually observed after several hours, further indicating that recycling of CHMP5/Vps60 polymers occurs in vivo, though Vps4 is seemingly not involved (Fig. 5 B and Fig. 6 F).

Strikingly, upon LLOMe treatment, CHMP4B and CHMP5 seemed to depict an alternative membrane coating on the damaged lysosomal membrane (Fig. 8 A), which is in good agreement with our in vitro observation of mutually exclusive binding of Snf7 and Vps60 polymers on SLBs (Fig. 2 F). Even

**Figure 6. Vps60 and Snf7 polymers behave differently in vivo.** Deletion of the *VPS60* gene reduces the number of ILVs per endosome in yeast. **(A–D)** Two-dimensional cross sections and three-dimensional models from 200-nm-thick section electron tomograms of wildtype (A and B) and *vps60Δ* yeast cells (C and D). Endosomal limiting membranes are depicted in yellow. Scale bar: 100 nm. **(E)** Number of ILVs counted per endosome lumen in each strain. Wildtype ROI = 12, *n* = 7; *vps60Δ*, ROI = 47, *n* = 4. P < 0.0001, two-tailed *t* test. **(F)** Western blots analyzing the distribution of Vps60-HA and Snf7 in detergent-solubilized membrane fractions resolved by rate-zonal density gradient centrifugation. Also indicated are the migrations of protein standards aldolase (158 kD), catalase (232 kD), and ferritin (440 kD). **(G)** Bar graphs representing the mean percentage of Vps60-HA and Snf7 in each gradient fraction examined in triplicate independent experiments; error bars indicate SD values. Source data are available for this figure: SourceData F6.

more interestingly, CHMP5 recruitment to nuclear envelope (NE) reformation during cell division was markedly absent in contrast to CHMP4B (Fig. 8 B, arrow) whereas both proteins are recruited during cytokinetic abscission (Fig. 8 B). Of note, a recent study reported absence of Did2 during NE sealing in fission yeast (Ader et al., 2022 *Preprint*), which might indicate that only a subset of ESCRT-III subunits is required for this function. These findings support the notion that CHMP4B and CHMP5 are involved in different ESCRT-III cellular functions.

Taken together, these observations of CHMP5 during ESCRT-III activity in vivo are consistent with our in vitro findings, indicating a distinct role for Vps60/CHMP5. First, CHMP5 showed delayed kinetics in the recruitment to the endosomes/lysosomes but also slower depolymerization compared with CHMP4B (Mercier et al., 2020). Moreover, the remarkable alternating of membrane domains of CHMP4B and CHMP5 polymers upon lysosomal recruitment and that CHMP5 is not involved in NE reformation further hints toward parallel but separate functions of Snf7/CHMP4B and Vps60/CHMP5 polymers.

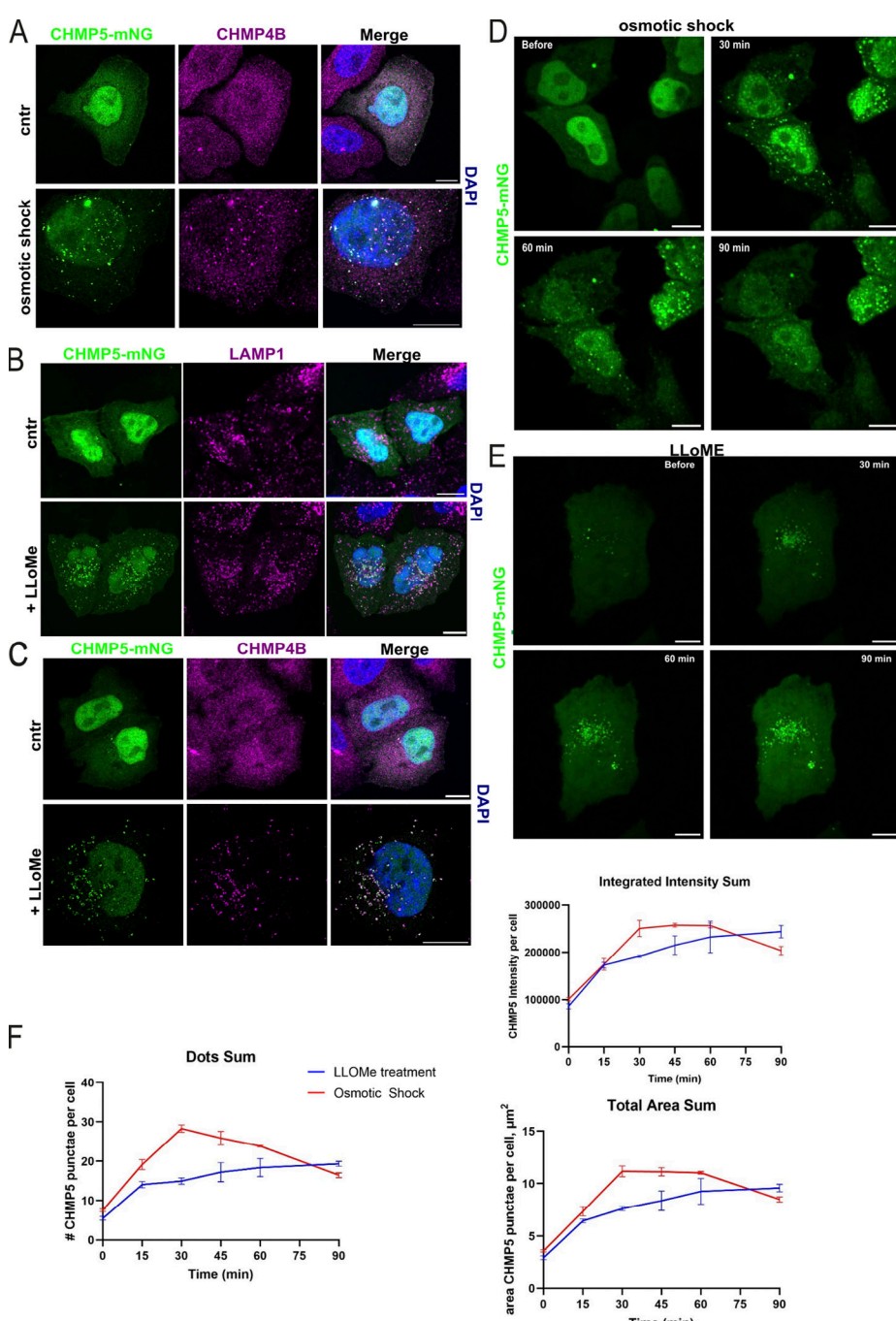

**Figure 7. CHMP5 activity in mammalian cells. (A)** Confocal images of immunofluorescence staining against CHMP4B of Hela-Kyoto cells expressing CHMP5-mNeonGreen under isosmotic or hypertonic conditions. **(B and C)** Confocal images of immunofluorescence staining against LAMP1 or CHMP4B of Hela-Kyoto cells expressing CHMP5-mNeonGreen with or without LLOMe treatment. Zoom-in of the deconvoluted image in C is shown in Fig. 8 A. **(D)** Live-cell imaging of Hela-Kyoto cells expressing CHMP5-mNeonGreen during osmotic shock. **(E)** Live-cell imaging of Hela-Kyoto cells expressing CHMP5-mNeonGreen upon LLOMe treatment. **(F)** Quantification of experiments described in D and E. Scale bar: 10 μm; D n = 3 ROI ≥ 400; E n = 3 ROI ≥ 2,500.

## Discussion

We here show that the ESCRT-III subunit Vps60 functions as the basis for a novel multisubunit ESCRT-III filament. We further propose this Vps60-based filament to potentially constitute the initiator of a second ESCRT-III polymerization sequence, an alternative to the Snf7-based sequence we recently unraveled (Fig. 8 C; Pfitzner et al., 2020). In detail, we found Vps60 to polymerize into ring-shaped or curled filaments on membranes, which, in analogy to the Snf7-based polymerization sequence, recruited ESCRT-III subunits Vps2, Vps24, Did2, and Ist1, before subsequently undergoing Vps4/Vta1-mediated filament turnover. Altogether, our results imply that Vps60, by acting as a template for initiating an alternative ESCRT-III filament, could functionally "replace" Snf7 in specific ESCRT-III functions that

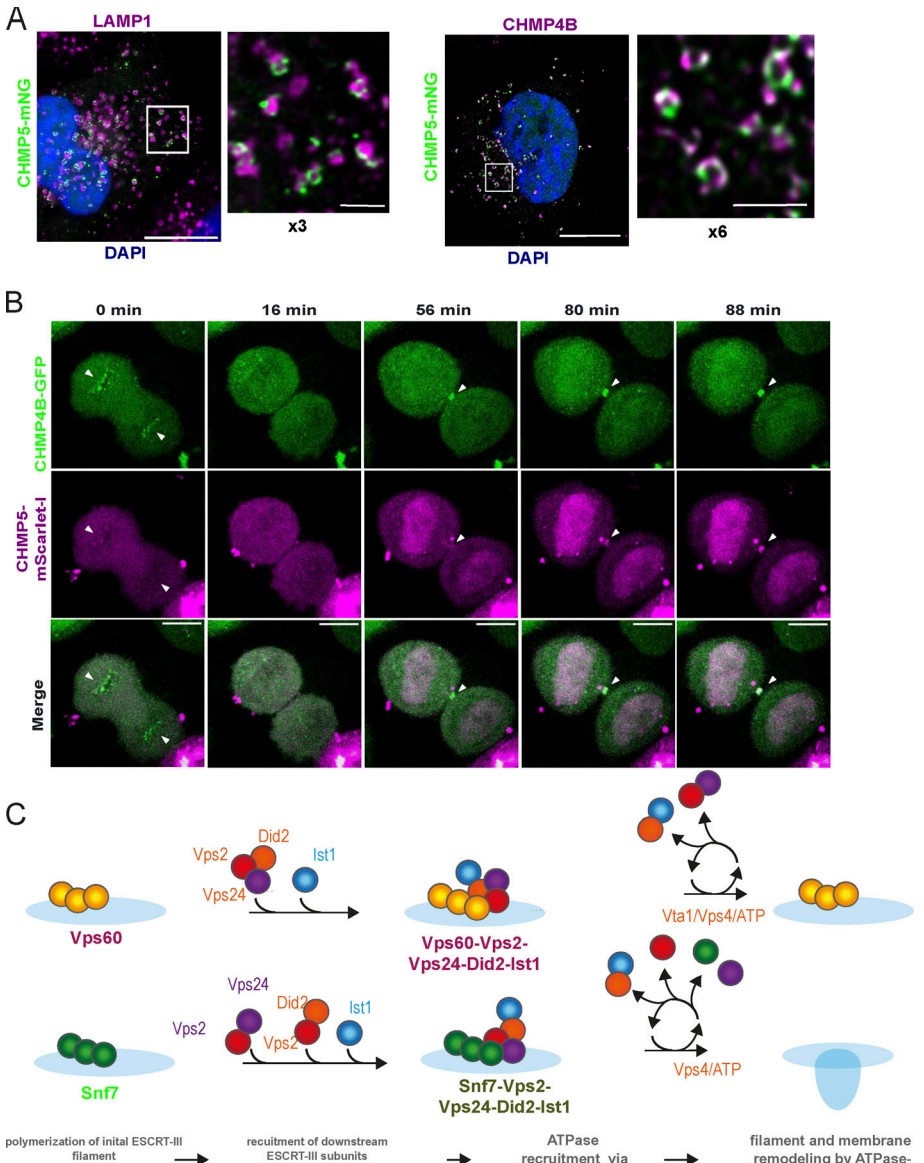

Figure 8. **CHMP5 and CHMP4B polymers have distinct dynamics in cells. (A)** Deconvoluted confocal images of immunofluorescence staining against LAMP1 or CHMP4B (same as Fig. 7 C, higher magnification) of Hela-Kyoto cells expressing CHMP5-mNeonGreen upon LLOMe treatment (scale bar: 10 μm, 2 μm inlet). **(B)** Live-cell imaging of Hela-Kyoto cells expressing CHMP4B-GFP and CHMP5-mScarlet-I undergoing nuclear envelop reformation and cytokinetic abscission (scale bar: 10 μm). Arrowheads indicate the reforming nuclear envelope and the cytokinetic bridge. **(C)** Cartoon of the proposed model for function of Vps60-based ESCRT-III filaments.

require biochemical properties only Vps60-initiated polymers provide (Teis et al., 2008; Adell et al., 2017; Babst et al., 2002; Henne et al., 2012), although it is also possible that Vps60-based filaments augment the function of Snf7-based filaments to achieve wildtype ESCRT-III activity.

In support of this notion, a recent study of ESCRT-III in *Plasmodium falciparum*–infected red blood cells suggests that PfVps32 (Snf7 homolog) and PfVps60 function in two parallel pathways during formation of extracellular vesicles (Avalos-Padilla et al., 2021). Likewise, we here find that ILV formation is partially inhibited in the absence of Vps60. The notion that Vps60 is part of the canonical Snf7-triggered ESCRT-III function was suggested in a recent study in yeast showing that recruitment of Vps60 to endosomal membrane was impaired in *snf7Δ*, *vps2Δ*, and *vps24Δ* mutants (Alsleben and Kölling, 2022). However, *vps60Δ* cells do not have the aberrant class E compartment endosomal morphology that is characteristic of *snf7Δ*, *vps2Δ*, and *vps24Δ* cells (Nickerson et al., 2010). Here, we find Snf7 and

Vps60 polymers to coexist independently on the same membrane in vitro and apparent segregated coating of endo-lysosomal membranes in vivo. Additionally, in agreement with Banjade et al. (2021), Vps60 polymerized spontaneously upon contact with membranes with a higher nucleation rate than Snf7 in our assays. Spontaneous nucleation of all established downstream ESCRT-III subunit (Vps2, Vps24, Did2, and Ist1) was absent in our assay. Thus, Vps60 might constitute an alternative ESCRT-III filament functioning in ILV function, although its interdependence with the canonical Snf7 pathway remains to be clarified, both in vivo and in vitro. Although our experiments exclude a direct exchange of both filaments mediated by ESCRT-III subunits, Vps4 or cofactors Vta1 and Bro1, Vps60 polymers could potentially function in a sequential manner together with Snf7 polymers during certain functions.

Overall, our observation of Vps2-Vps24-Did2 recruitment to Vps60 filament is strongly supported by a recent analysis of ESCRT-III complexes isolated from yeast which reports

interactions of Vps60 (Mos10) with Did2, Vps2, and Vps24 (Alsleben and Kölling, 2022). Vps4-triggered filament remodeling has previously been established as one of the key factors of ESCRT-III activity in cells (Adell et al., 2017; Guizetti et al., 2011; Mierzwa et al., 2017). In our study, Vps60-based filaments were able to recruit Vps4 and underwent partial disassembly. Upon addition of the Vps4 activator Vta1, a disassembly hierarchy between the various subunits was established that is similar to sequential depolymerization observed in Snf7-based filaments (Pfitzner et al., 2020; Mierzwa et al., 2017). In detail, Vps24 and Vps2 were depolymerized first, followed by Did2 and Ist1. This delay in Did2 disassembly could potentially be mediated by its direct interaction with Vta1 (Yang et al., 2012; Shen et al., 2016; Nickerson et al., 2010; Monroe et al., 2017; Norgan et al., 2013; Azmi et al., 2008). Alternatively, by globally stimulating Vps4 activity, Vta1 might emphasize pre-existing biases in subunit susceptibility to Vps4 e.g., their accessibility within filaments. In stark contrast to Snf7-based polymers, Vps60 polymers are not disassembled in any of the conditions we tested. In coherence, no accumulation of high-molecular-weight Vps60 polymer, as seen for Snf7, was observed upon disruption of Vps4 activity in cells. In contrast, Vps60 filaments underwent gross depolymerization in these conditions. These results strongly argue for a Vps4-independent recycling mechanism for Vps60 polymers in cells. Vp60-based filaments might be targeted by another microtubule interacting and trafficking domain containing AAA-ATPase besides Vps4.

Overall, the step-wise subunit recruitment and sequential depolymerization of Vps60-based polymers bear clear resemblance to the Snf7-based polymerization sequence we previously described (Pfitzner et al., 2020) and suggests the Vps60-based filament described in this study might form the initiation of an alternative ESCRT-III polymerization sequence (Fig. 8 C). Bioinformatic analysis of ESCRT-III proteins classifies the subunits into two groups according to their domain conservation: the Snf7 family, encompassing Vps20, Snf7, and Vps60 (class I), groups the nucleators and initiators of ESCRT-III (class I), and the Vps2 family, encompassing Vps2, Vps24, and Did2 (class II), groups the "pivots" that recruit Vps4 (Leung et al., 2008). At a closer look, both Vps60-based and Snf7-based ESCRT-III polymers and their corresponding presumed polymerization sequences share remarkable similarities regarding their organization, indicating that ESCRT-III–mediated membrane remodeling might follow a general mechanism (Fig. 8 C). An initial class I polymer (ring or spiral) mediates binding of class II subunits (helices), which then recruit an ATPase to trigger filament remodeling to promote ESCRT-III activity. In fact, this generalized progression of events is supported by the observation that direct recruitment of Vps4 via an initiating class I protein is non-functional in vivo (Banjade et al., 2021), potentially due to premature disassembly of the ESCRT-III polymer (Banjade et al., 2021) or a lack of filament remodeling capacity (Pfitzner et al., 2021).

A remaining, yet essential question about the Vps60-based dynamic ESCRT-III polymers is their cellular function. Overall, the specific differences, such as structure and kinetics between both ESCRT-III polymers and their associated sequences, hint toward functional specialization of the filaments, adopting

requirements of distinct cellular functions. This notion is supported by the different dynamics for the recruitment and depolymerization of CHMP5 compared with CHMP4B, the lack of CHMP5 recruitment during NE reformation, the unique cellular localization of GFP-tagged Vps60 in vivo to not only endosomes, as observed for Snf7, but also the vacuolar membrane (Banjade et al., 2021; Alsleben and Kölling, 2022), and by the separate functional pathways that were suggested for ESCRT-III subunits PfVps32 (Snf7 homolog) and PfVps60 during extracellular vesicle formation in *P. falciparum*–infected red blood cells (Avalos-Padilla et al., 2021).

## Materials and methods

### Protein purification
Vps60 (pGEX4T-Vps60) and Vta1 (pGEX4T-Vta1; gift from David Katzmann lab, Mayo Clinic, USA) were expressed for 4 h in Bl21 *Escherichia coli* following IPTG induction. Bacteria were lysed (1% triton, 20 mM Hepes, pH 8.0, 150 mM NaCl, cOmplete) using sonication and soluble lysate was loaded on GST-resin (washing buffer: 20 mM Hepes, pH 8.0, 150 NaCl) before on-column cleavage using tobacco etch virus protease (TEV) protease was performed. Vta1-GST was eluted prior to TEV treatment as on-column digest was inefficient (elution buffer: 150 mM NaCl, 10 mM glutathione, 20 mM Hepes, pH 8.0). TEV protease was removed using nickel-nitriloacetic acid resin and the purified protein of interest was dialyzed against storage buffer (20 mM Hepes, pH 8.0) before concentration (Fig. S1 A).

Increase of Vps4 ATPase activity upon addition of Vta1 was checked by malachite green assay at the end of each purification to confirm the proper functioning of Vta1. In short, a malachite green stock solution was prepared by mixing three parts of 0.045% malachite green in water with two parts of 4.2% ammonium molybdate in 4 M HCl for 1 h under constant stirring before supplementation with 0.01% Tween20. For quantification, a calorimetric measurement at 620 nm was performed 5 min after mixing three parts of malachite green stock solution with one part of reaction solution either resulting from incubation of Vps4 and Vta1in reaction buffer (20 mM Hepes, 150 mM NaCl, 2 mM MgCl2, pH = 7.5) supplemented with 1 mM ATP or a phosphate calibration curve.

Following the labeling procedure given by the reagent provider, Snf7, Vps2, Ist1, and Vps24 were labeled with TFP-AlexaFluor-488 (Ref N°A-30005; Thermo Fisher Scientific). Vps2 and Vps60 were labeled with TFP-Atto-565 (Atto-Tec AD 565-3). Did2 was labeled with maleimide-AlexFluor-488 (A-30005; Thermo Fisher Scientific). Vps2 was labeled with NHS-Atto-647N (Atto-Tec AD 647N). If not otherwise mentioned, the following protein concentrations were used: ESCRT-II 1 µM, Vps20 1 µM, Bro1 500 nM, Vps60 50 nM, Snf7 400 nM, Vps2 1 µM, Vps24 1 µM, Did2 1 µM, Ist1 1 µM, Vps4 1 µM, Vta1 500 nM, and ATP 2 mM. In general, labeled proteins were mixed 1:1 with unlabeled proteins.

### Preparation of GUVs and LUVs
GUVs were prepared by electroformation: 20–30 µl of a 2 mg/ml lipid solution in chloroform (DOPC:DOPS:DOPE-Atto647N:

DSPE-PEG [2000] Biotin, 6:4:0.01:0.003; Avanti Polar Lipids; Atto-tec) was dried on indium-tin oxide (ITO)–coated glass slides for 1 h. For experiments including Atto647N-Vps2, unlabeled GUVs were used. A growth chamber was assembled by clamping a rubber ring between the ITO slides and filled with 500 µl of a sucrose buffer osmotically equilibrated with the experimental buffer. ITO slides were then connected to an AC generator set under 1V AC (10 Hz) for 1.5 h. GUVs were stored at 4°C for at maximum a week.

For LUV preparation, DOPC:DOPS (6:4; 10 mg/ml) mixture was evaporated in a glass tube, 500 µl of buffer was added, and the tube was vortexed followed by five times freezing and thawing. LUVs were stored at −20°C and extruded with a 200-nm filter before use.

## Supported membrane bilayer assay

Supported membrane bilayer assay was performed as described in Chiaruttini et al. (2015). Experiments were performed in 20 mM Tris, pH 6.8, 200 mM NaCl, and 1 mM $MgCl_2$. 2 mM DTT was added to the buffer for experiments including Ist1. GUVs diluted in buffer were burst on a plasma-cleaned coverslip forming the bottom of a flow chamber (coverslip and sticky-Slide VI 0.4; Ibidi) to form supported bilayers. Thereafter, the chamber was passivated with casein (1 mg/ml; Sigma-Aldrich) for 10 min and washed with buffer before the experiment was conducted. Subsequent changes of protein or buffer solutions in the chamber were made via a syringe pump connected to the flow chamber. Briefly, protein preparations are diluted in reaction buffer with 80 µl final volume. In presence of Ist and Did2, 0.8 µl DTT 1 mM was added, and reaction buffer had 2 mM $MgCl_2$ added. All proteins have a final concentration of 1 µM during acquisitions except Vps60 and Snf7. Snf7 patches are pregrown at 400 nM untilSnf7 is washed out before other protein is added. Vps60 was tested with many concentrations for dynamics assays and added at 50 nM for 9 min when pregrown for experiments.

Partially adhered vesicles were prepared as described in Chiaruttini et al. (2015). Briefly, a flow chamber was assembled from a coverslip and sticky-Slide VI 0.4, and Ibidi was incubated with Avidin (0.1 mg/ml) for 10 min before washing with buffer (20 mM Tris, pH 6.8, 200 mM NaCl, and 1 mM $MgCl_2$), this before adding GUVs (including 0.03% DSPE-PEG [2000] Biotin) diluted in buffer. As soon as GUVs started to attach, biotinylated albumin (1 mg/ml; Sigma-Aldrich) was added to stop attachment and prevent bursting of the GUVs.

## Image acquisition

Confocal imaging was performed on an inverted spinning disc microscope assembled by 3i (Intelligent Imaging Innovation) consisting of a Nikon base (Eclipse C1; Nikon), a 100× 1.49 NA oil immersion objective, and an EVOLVE EM-CCD camera (Ropper Scientific Inc.). For analysis of supported bilayer experiments, 3-µm-thick Z-stacks were maximally projected using a Fiji plugin. X-y drift of the microscopy was corrected using the plugin Turboreg and a custom-written ImageJ macro. For artificial membrane neck experiments, 15-µm-thick Z-stacks were acquired.

## EM

For EM experiments, LUVs were diluted 1:100 in buffer (20 mM Tris, pH 6.8, 200 mM NaCl, and 1 mM $MgCl_2$), spun down (10 min, 5,000 $g$), and resuspended in 250 nM Vps60 for 1 h at 4°C. Samples were absorbed onto carbon-coated grids Cu 300 and stained with 2% uranyl acetate for 30 s. Images were acquired on a Tecnai G2 Sphera (FEI) electron microscope. Cryofixation of cells, electron tomography, and three-dimensional modeling have been described previously (Nickerson et al., 2010). Log-phase yeast cells were high-pressure frozen and freeze-substituted at −90°C until fixed (Winey et al., 1995; Giddings, 2003) using a Leica automated freeze-substitution system. Standard media for tomography and immunolabel was 0.1% uranyl acetate and 0.25% glutaraldehyde in anhydrous acetone. An additional 2% glutaraldehyde was sometimes used to ensure morphological preservation. In brief, samples were warmed in the Leica automated freeze-substitution system from liquid nitrogen temperatures to −90°C and incubated for 3 d. Samples were washed in additional anhydrous acetone for a day while warmed to −50°C and then embedded in Lowicryl HM20 over a period of 3 d. Embedded samples were polymerized under UV radiation at −50°C and slowly warmed to room temperature over 4 d. Plastic blocks were sectioned (150–300 nm) on a Leica Ultramicrotome and placed on rhodium-plated copper slot grids with Formvar films. Grids were labeled on both sides with fiducial 15-nm nanogold. Dual tilt series images were collected from −60 to +60° tilt range with 1° increments (Mastronarde, 1997) at 200 kV by using a Tecnai 20 FEG microscope (FEI). Tilt series were imaged at 29,000× with a 0.77-nm pixel (binning 2). IMOD and 3dmod software from the 3DEM lab (Kremer et al., 1996) were used for tomogram generation and modeling, respectively. Digital images were processed using SlideBook (Intelligent Imaging Innovations) and Photoshop 7.0 (Adobe Systems). Statistical analyses were performed using Prism 4.0 (GraphPad Software).

## Optical tweezer tube pulling experiment

Membrane nanotube pulling experiments were performed on the setup published in Chiaruttini et al. (2015) allowing simultaneous optical tweezer application, spinning disc confocal, and brightfield imaging based on an inverted Nikon eclipse Ti microscope, and a 5-W 1,064-nm laser focused through a 100 × 1.3 NA oil objective (ML5-CW-P-TKS-OTS; Manlight). Membrane nanotubes were pulled with streptavidin beads (3.05 µm; Spherotec) from a GUV containing 0.01% DSPE-PEG (2000) Biotin and aspired in a motorized micropipette (MP-285; Sutter Instrument). Proteins were injected using a slightly bigger micropipette connected to a pressure control system (MFCS-VAC -69 mbar; Fluigent).

## Rate-zonal density gradient analysis of ESCRT-III

Gradient analysis of Vps60-HA and Snf7 polymer size and abundance was performed as described in Johnson et al. (2017). Briefly, 20–30 $OD_{600}$ units of yeast cells were converted to spheroplasts and osmotically lysed in ice-cold PBS lysis buffer containing 0.5% Tween-20. The cells were then homogenized on ice and the membrane fraction was pelleted by

centrifugation at 16,000× $g$ at 4°C, then resuspended in lysis buffer, and passed through a 25-gauge needle five times before loading on top of a linearized glycerol gradient (10–40%) prepared in PBS and 0.5% Tween20. The gradient was centrifuged at 100,000× $g$ for 4 h at 4°C. 1-ml fractions were collected and 10% trichloroacetic acid was added to precipitate the proteins on ice. Precipitates were pelleted by centrifugation at 16,000× $g$ and washed twice by sonication in ice-cold acetone. Pellets were dried, resuspended in 50 µl of Laemmli buffer, and boiled for 5 min. 15 µl of each gradient protein sample was resolved by SDS-PAGE, transferred to a nitrocellulose membrane, and Western blotting was performed using anti-HA monoclonal antibodies (clone 12CA5, 11583816001; Roche) versus rabbit anti-Snf7 antiserum (Babst et al., 1998) using secondary antibodies that were peroxidase conjugate-goat anti-mouse (product no. A4416) or peroxidase-conjugate goat anti-rabbit (product no. A6154) from Sigma-Aldrich. Western blot imaging and quantitation were performed using the BioRad Chemidoc MP imaging system and the Image Lab 5.0 software. The amounts of Vps60 and Snf7 were quantitated in triplicate experiments using BioRad software, and the mean values of protein detected in each lane were plotted along with standard deviations.

## Cell culture, stable cell lines, and reagents
Human HeLa "Kyoto" cells were maintained in DMEM (Gibco) medium, supplemented with 10% fetal bovine medium, 100 U/ml penicillin, and 100 µg/ml streptomycin. Cells were maintained at 37°C supplemented with 5% $CO_2$. A stable HeLa cell line expressing CHMP4B-eGFP was obtained from Anthony A. Hyman (Max Planck Institute of Molecular Cell Biology and Genetics, Dresden, Germany) and was maintained with the media described before plus G418 (geneticin) 300 µg/ml. These cells were routinely tested mycoplasma-negative by GATC Biotech.

The lysomotropic drug LLOMe was purchased from Sigma-Aldrich (L7393). LLOMe was dissolved in DMSO and properly aliquoted and stored at −20°C.

For transient expression experiments, Lipofectamine 3000 from Thermo Fisher Scientific was used for live-imaging and immunofluorescence experiments, and TransIT-LT1 Transfection Reagent (MIR2300) from Mirus was used for quantification experiments.

## Plasmid constructs
Human CHMP5 protein (Q9NZZ3 Uniprot Accession Number) was synthesized de novo and subcloned using NheI and XhoI restriction enzymes into the pcDNA3.1 vector for transient expression in HeLa cells. This vector had already the fluorescent tags mNeonGreen and mScarlet-I, also synthesized de novo with optimized codons for human cell expression. A long linker inspired by Long Affinity Purified linker was designed to fuse the protein with the tag.

## Antibodies
Rabbit monoclonal antibody against LAMP1 (1/1,000 dilution for immunofluorescence) was from Cell Signaling (9091), and rabbit

polyclonal antibody against CHMP4B (1/300) was from Proteintech (13683-1-AP). All secondary antibodies used for immunofluorescence studies were bought from Invitrogen (Thermo Fisher Scientific).

## Live-imaging, immunofluorescence, and quantification experiments
For live-cell imaging, $2.5 \times 10^5$ cells were seeded into 35-mm MatTek glass-bottom microwell dishes. Cells were analyzed by a Nikon Ti spinning disc microscope and a 60× Plan-Apochromat (1.40 NA) lens. For image acquisition, cells were rinsed three times with 1 ml of FluoroBrite DMEM (Gibco) medium and then cells were maintained during imaging at 37°C and 5% $CO_2$. Images obtained were analyzed in Fiji/ImageJ. Osmotic shocks were carried out as described in Mercier et al. (2020).

For the immunofluorescence studies, $5 \times 10^4$ cells were seeded on 1.5 round glass coverslips (Electron Microscopy Sciences) in 24-well plates at 37°C and $CO_2$ for 24 h. Cells were fixed in 4% PFA in phosphate-buffered saline (PBS) for 15 min. Cells were washed twice with PBS and once with glycine 0.1 M to block unreacted aldehydes. Then, cells were blocked and permeabilized with 3% BSA + 0.2% saponin for 1 h. Primary and secondary antibodies were incubated for 1 h with 1% BSA + 0.2% saponin, and cells were washed with PBS containing 0.2% saponin. Finally, they were mounted using Fluoroshield with DAPI (GeneTex) and sealed using nail polish (Electron Microscopy Sciences). Subsequently, cells were analyzed by a confocal microscope (Leica SP8 LIGHTNING) and deconvolved using LAS X software.

For quantification analyses, cells were seeded in 96-well plates and transformed using TransIT-LT1 Transfection Reagent. Transfected cells were analyzed with a confocal automated microscopy (Molecular Device). Analysis of these experiments was done via MetaXpress Custom Module editor software. The images were segmented to choose those cells with appropriate level of expression, and relevant masks were generated to extract the relevant measurements. To facilitate segmentation of ESCRT endosomes/lysosomes, the top hat deconvolution method was applied to reduce the background noise and highlight bright dots.

## Quantification and statistical analysis
For quantification of supported bilayer experiments, the integrated fluorescence intensity of membrane patches (Vps60) of single proteins patches (Snf7) was measured using Fiji, background at time 0 min subtracted, normalized to time point 0, and a kymograph was extracted for dynamic experiments. Fluorescence intensities were normalized by their maximum value. To determine the colocalization of Atto565-Vps60 and Alexa488-Snf7 or Atto647-Vps2 and Atto565-Vps60 or Alexa488-Snf7, relative fluorescence was measured along linearized membrane contours and relative fluorescence values were binarized (1 above threshold, 0 below, thresholds: 0.3 Snf7, 0.4 DOPE). The percentage of no colocalization was extracted by the proportion of pixels with the value 1 from the Snf7 channel for which the value in the membrane channel was 0. No colocalization was only counted at a minimal distance of four pixels to the nearest membrane neck (value 1).

For quantification of nucleation rates, membrane areas were isolated from images, and Vps60-puncta or total fluorescence of Vps60 after background substraction ($t$ = 0 min) was extracted and divided by the total membrane area.

For quantification of the number of ILVs, wildtype and mutant yeast were high-pressure frozen and fixed in parallel for each experiment. 100 cell profiles were sampled in 80-nm-thin sections imaged by transmission EM ($n$ = 100), and 5–10 representative cell profiles were sampled by electron tomography of 200-nm-thick sections ($n$ = 5–10). Between one and five organelle structures were sampled per thick section at 20,000× magnification. The means and standard error of multivesicular bodies (MVB) and ILV diameters were determined. Data distribution was assumed to be normal but this was not formally tested.

For all experiments, the mean and standard deviation were calculated. The number of independent experiments ($n$) and the number of events (ROI) analyzed are indicated in the corresponding figure legends. The graph and statistics were done using Prism 8 (GraphPad software).

### Online supplemental material
Fig. S1 shows the analysis of the polymerization behavior of ESCRT-III subunits under different conditions and serves as a complementation to data presented in Fig. 1, A and B. Fig. S2 shows the mutually exclusive membrane binding of Vps60 and Snf7 and serves as a complementation to data presented in Fig. 2, D–I. Fig. S3, A–D, shows the mutually exclusive membrane binding of Vps60 and Snf7 and serves as a complementation to data presented in Fig. 2, D–I. Fig. S3, E–G, shows representative images of Vps60-based ESCRT-III copolymers and resulting membrane deformations in complementation to data presented in Fig. 4. Fig. S4 shows the membrane recruitment of downstream ESCRT-III subunits by Vps60-based polymers and serves as complementation to data presented in Fig. 3. Fig S5, A–D, shows effect of Vta1 on the turnover and depolymerization of Did2 and Ist1 from Vps60-based ESCRT-III polymers. Fig. S5, F and G, shows the independent ESCRT-III subunit recruitment to Snf7 and Vps60 polymers in complementation to data presented in Fig. 5, H–K. Video 1 shows a representative example of the tomographic analysis of wildtype yeast as described in Fig. 7, A and B. Video 2 shows a representative example of the tomographic analysis of vps60Δ yeast as described in Fig. 7, C and D.

### Data availability
All data generated or analyzed during this study are included in this published article (and its supplementary information files). Raw image data are available at https://doi.org/10.5281/zenodo.7963500.

### Acknowledgments
The authors want to thank the National Centre of Competence in Research Chemical Biology for constant support during this project. We thank the ACCESS automated microscopy, high-content, and high-throughput screening facility and the bio-imaging core facility of the University of Geneva for constant support.

A. Roux acknowledges funding from the Swiss National Fund for Research Grants No. 31003A_149975, No. 31003A_173087, and No. 31003A_200793, and the European Research Council Synergy Grant No. 951324_R2-TENSION. G. Odorizzi acknowledges funding from the National Institutes of Health/National Institute of General Medical Sciences, grant number GM111335. C. Bernat-Silvestre acknowledges a post-doctoral fellowship from the Fundación Alfonso Martín Escudero. T. Peltier acknowledges training grant support from the National Institutes of Health/National Institute of General Medical Sciences, grant number T32GZM142607. Open Access funding provided by Université de Genève.

Authors contributions: A. Roux and A.-K. Pfitzner conceptualized the study. A.-K. Pfitzner and H. Zivkovic performed and analyzed in vitro and negative stain EM experiments. C. Bernat-Silvestre performed and analyzed CHMP5 experiments. M. West performed and analyzed EM tomography experiments and T. Peltier the assembly-disassembly analysis by density gradients, both supervised by G. Odorizzi. F. Humbert and H. Zivkovic purified proteins. A.-K. Pfitzner and A. Roux wrote the manuscript with help from H. Zivkovic.

Disclosures: The authors declare no competing interests exist.

Submitted: 8 June 2022

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

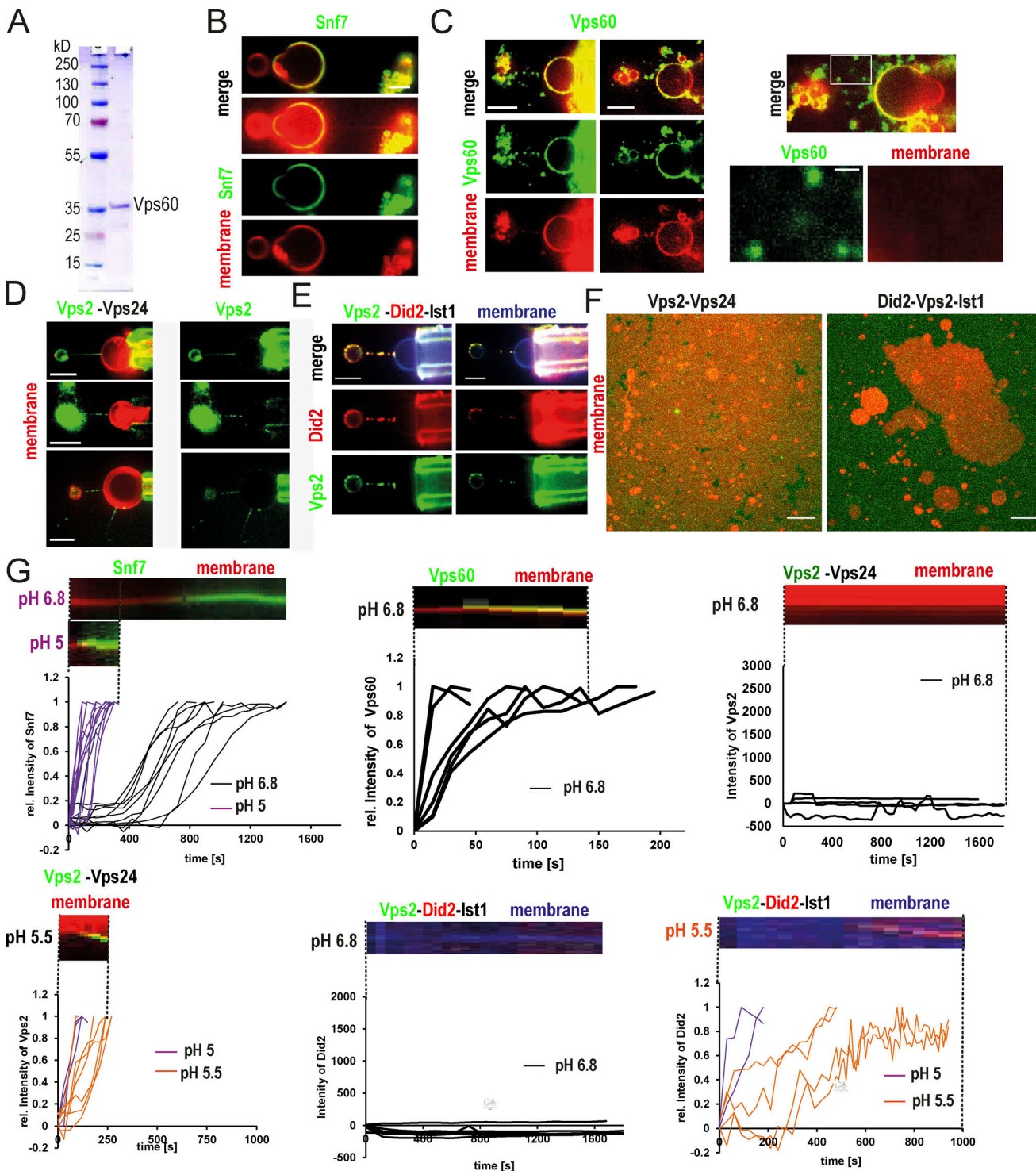

Figure S1. **Characterization of ESCRT-III protein properties. (A)** Coomassie gel of purified Vps60. **(B)** Single-channel confocal microscopy images of the experiment described in Fig. 1 A. Snf7-Alexa488 (green), membrane nanotubes (red); scale bar: 5 µm. **(C)** Confocal microscopy images of Vps60 (green) binding to membrane nanotubes (red; scale bar: 5 µm). **(D)** Confocal microscopy images of Vps2-Alex488 (green) and Vps24 binding to membrane nanotubes (red; scale bar: 5 µm). **(E)** Confocal microscopy images of Vps2-Alex488 (green), Did2-Atto565 (red), and Ist1 binding to membrane nanotubes (blue; scale bar: 5 µm). **(F)** Confocal images of (Vps2-Alexa488)-Vps24 (green) or (Did2-Alexa488)-Vps2-Ist1 (green) nucleation on SLBs (red; scale bar: 10 µm). **(G)** Quantification and kymographs of dynamics of Snf7 (A), Vps60 (B), Vps2-Vps24 (C), or Vps2-Did2-Ist1 (D) binding to membrane at indicated pH. Source data are available for this figure: SourceData FS1.

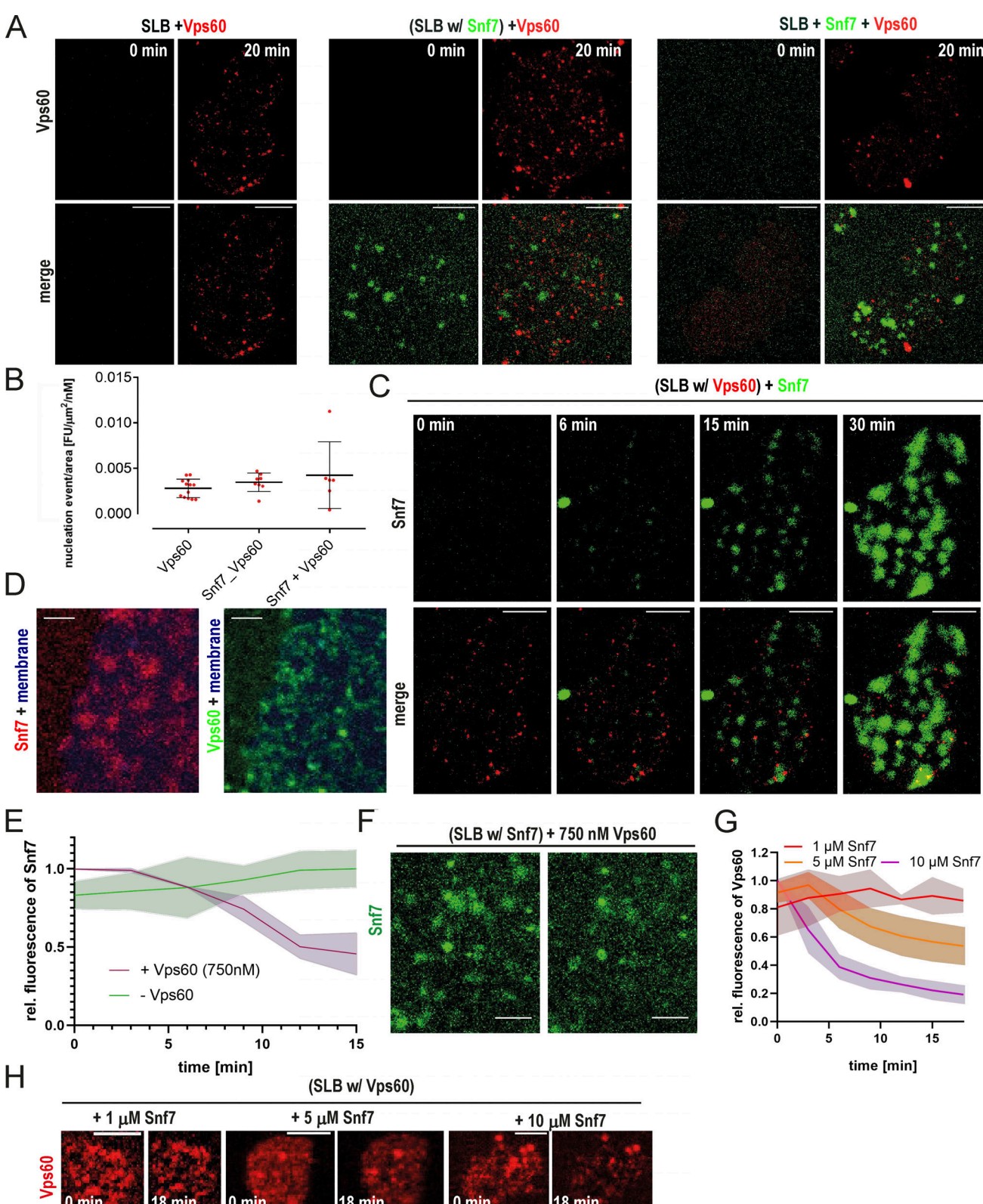

Figure S2. **Vps60 nucleation pattern on SLBs. (A)** Confocal images of time-lapse experiments of SLBs incubated with the indicated mixture of proteins (scale bar: 10 μm) described in Fig. 2 D. **(B)** Quantification of Vps60 intensity from experiments described in Fig. 2 A ($n \geq 3$, Vps60 ROI = 13, Snf7_Vps60 ROI = 8, Snf7+Vps60 ROI = 6; scale bar: 10 μm). **(C)** Confocal images of time-lapse experiments of the addition of Alexa488-Snf7 to Vps60-covered SLBs (scale bar: 10 μm). **(D)** Confocal images of experiments described in Fig. 2 F (scale bar: 2 μm). **(E and F)** Quantification of Snf7 intensity (E) and confocal images (F) of time-lapse experiments of addition of 750 nM Vps60 (unlabeled) to SLBs with pregrown Snf7 patches (scale bar: 5 μm). **(G and H)** Quantification of Vps60 intensity (G) and confocal images (H) of time-lapse experiments of the addition of indicated concentrations of Snf7 (unlabeled) to SLBs with preincubated with Atto565-Vps60 (scale bar: 2 μm).

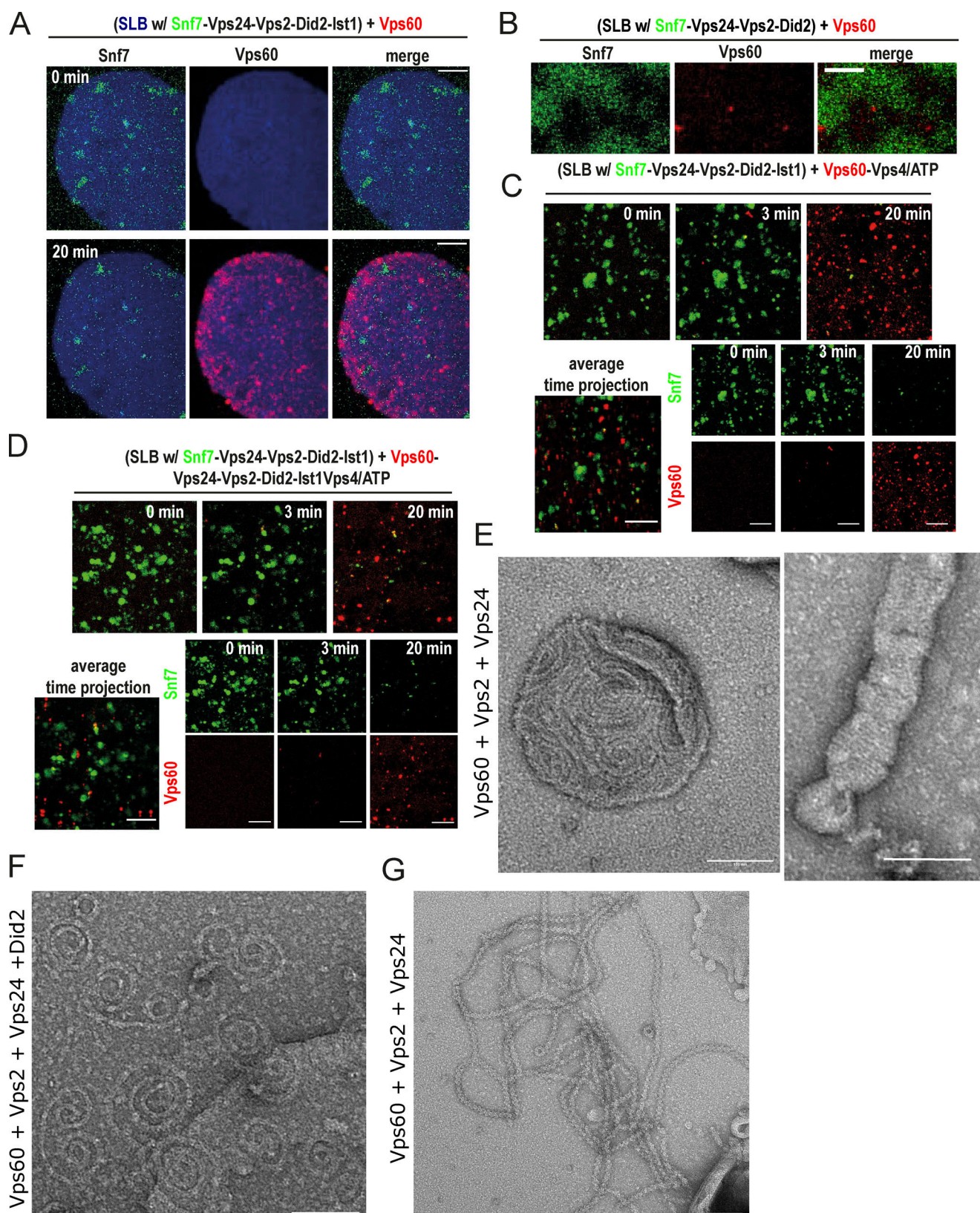

Figure S3. **Vps60 and Snf7 bind membrane independently from each other. (A)** Confocal images of experiment described in Fig. 2 H (scale bar: 10 µm). **(B)** Confocal images of the time-lapse experiment of addition of Vps60 (red) to SLBs (blue) preincubated with Snf7 (green), Vps2, Vps24, and Did2 (*n* = 3; scale bar: 2 µm). **(C and D)** Confocal images and average time projection of time-lapse experiments of the addition of Atto565-Vps60 and Vps4/ATP (C) or Vps24, Vps2, Did2, Ist1, and Vps4/ATP (D; *n* ≥ 3, ROI ≥ 12; scale bar: 5 µm). **(E and F)** Negative stain electron micrographs of Vps60 filaments incubated with indicated proteins polymerized on LUVs (scale bar: 100 nm). **(G)** Negative stain electron micrographs of Vps60 filaments incubated Vps2 and Vps24 (scale bar: 100 nm).

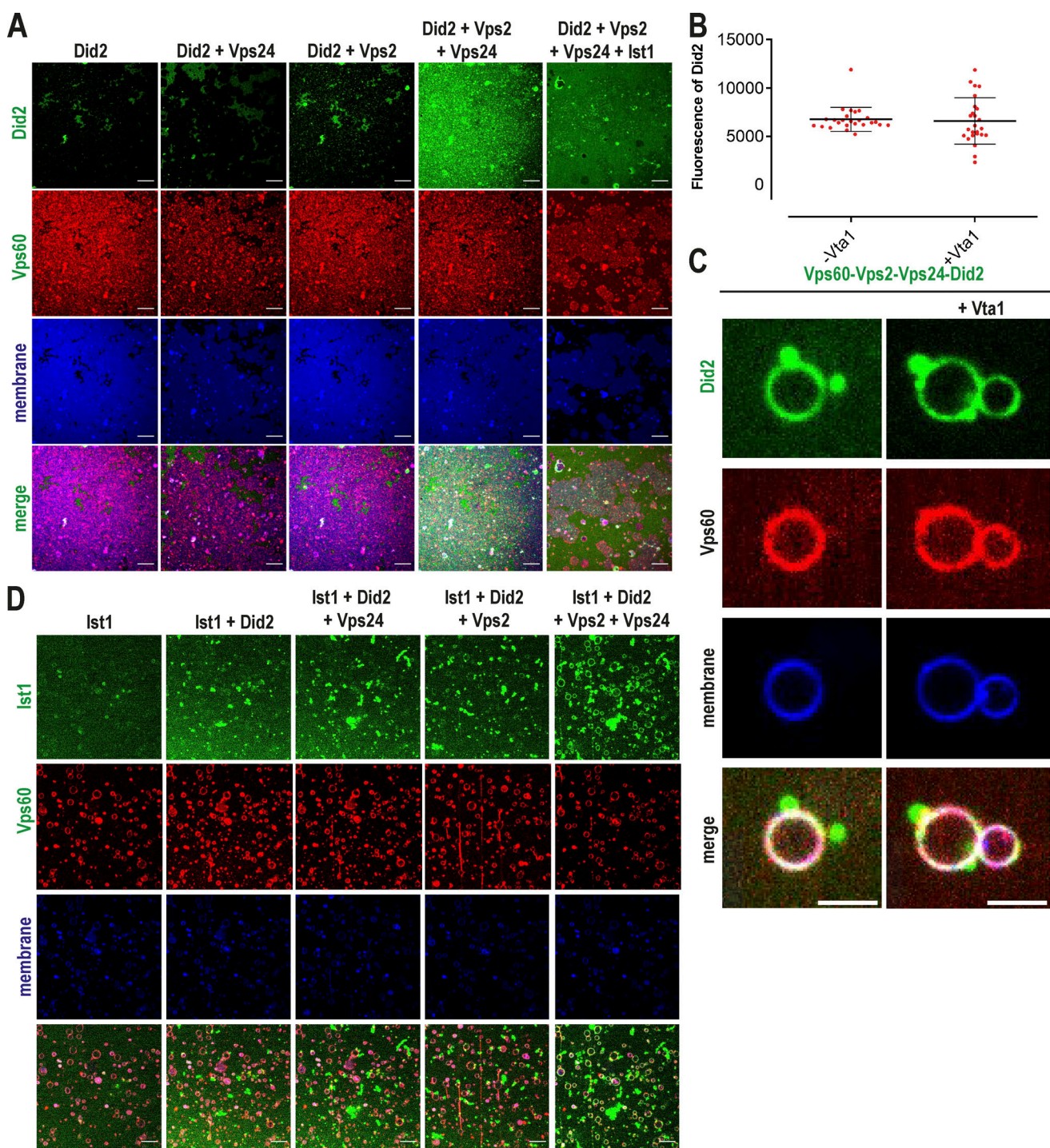

Figure S4. **Membrane-bound Vps60 recruits ESCRT-III proteins. (A)** Confocal images of experiments described in Fig. 3 E. **(B and C)** Quantification of Did2 intensity (B *n* = 3, −Vta1 ROI = 24, +Vta1 ROI = 25) and confocal images (C) of Vps60-coverd SLBs incubated with Vps2, Vps24, Alexa488-Did2, and Ist1 in presence or absence of Vta1. **(D)** Confocal images of experiments described in Fig. 3 F. Scale bar: 10 μm.

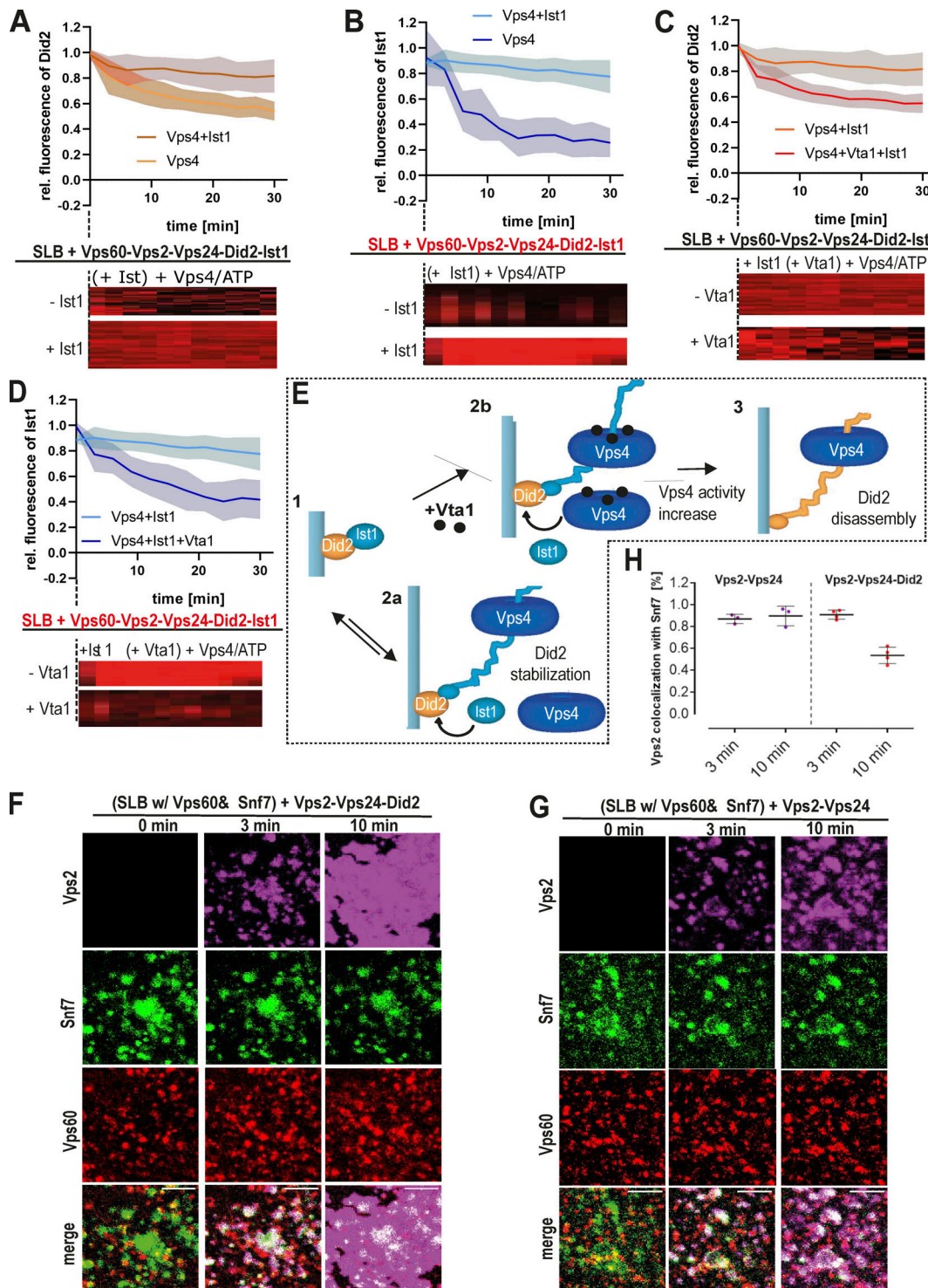

Figure S5. **Vps4-triggered depolymerization of Did2 and Ist1 from Vps60-based ESCRT-III filaments. (A)** Quantification of Did2 intensity and kymographs of time-lapse experiments of the addition of Vps4/ATP or Ist1 and Vps4/ATP to SLBs preincubated with Vps60, Vps2, Vps24, Alexa488-Did2, and Ist1. Related to Fig. 5 E (−Ist1 condition) and panel C (+Ist1 corresponds to −Vta1 experiment). **(B)** Quantification of Ist1 intensity and kymographs of time-lapse experiments of addition of Vps4/ATP or Ist1 and Vps4/ATP to SLBs preincubated with Vps60, Vps2, Vps24, Did2, and Alexa488-Ist1. Related to panel D (+Ist1 corresponds to −Vta1 experiment). **(C)** Quantification of Did2 intensity and kymographs of time-lapse experiments of the addition of Ist1 and Vps4/ATP or Ist1, Vps4/ATP, or Vta1 to SLBs preincubated with Vps60, Vps2, Vps24, Alexa488-Did2, and Ist1. Related to A (+Ist1 corresponds to −Vta1 experiment). **(D)** Quantification of Ist1 intensity and kymographs of time-lapse experiments of addition of Ist1 and Vps4/ATP or Ist1, Vps4/ATP, and Vta1 to SLBs preincubated with Vps60, Vps2, Vps24, Did2, and Alexa488-Ist1. Related to B (+Ist1 corresponds to −Vta1 experiment). **(E)** Cartoon of the model for interplay of Vps4, Did2, and Ist1 during disassembly of ESCRT-III polymers in presence or absence of Vta1. **(F)** Single-channel confocal images of the experiment shown in Fig. 5. **(G)** Single-channel confocal images of the experiment shown in Fig. 5. **(H)** Quantification of Vps2 fluorescence intensities total membrane coverage and colocalization with Snf7 patches from experiments described in Fig. 5, H and I ($n \geq 3$). Scale bar: 10 µm.

Video 1.  **Representative example of the tomographic analysis of wildtype yeast as described in** Fig. 6, A and B**.** Z series from 200-nm-thick section electron tomograms and 3D model of two MVBs from wildtype cells. Endosomal limiting membranes are depicted in yellow. ILV are depicted in red. Scale bar: 100 nm. Playback speed: accelerated 20×.

Video 2.  **Representative example of the tomographic analysis of ΔVps60 yeast as described in** Fig. 6, C and D**.** Z series from 200-nm-thick section electron tomograms and 3D model of multiple MVBs from ΔVps60 cells. Endosomal limiting membranes are depicted in yellow. ILV are depicted in red. Scale bar: 100 nm. Playback speed: accelerated 20×.

