## [Peer Review File · The Journal of Cell Biology]

Vps60 initiates alternative ESCRT-III filaments

Anna-Katharina Pfitzner, Henry Zivkovic, César Bernat-Silvestre, Matt West, Tanner Peltier, Frédéric Humbert, Greg Odorizzi, and Aurélien Roux

Corresponding Author(s): Aurélien Roux, University of Geneva and Anna-Katharina Pfitzner, University of Geneva

Review Timeline:

Submission Date:	2022-06-08
Editorial Decision:	2022-07-22
Revision Received:	2023-03-08
Editorial Decision:	2023-05-02
Revision Received:	2023-05-25

Monitoring Editor: Pier Paolo Di Fiore

Scientific Editor: Tim Spencer

Transaction Report:

DOI: <https://doi.org/10.1083/jcb.202206028>

July 22, 2022

Re: JCB manuscript #202206028

Dr. Aurelien Roux
University of Geneva
Biochemistry Department University of Geneva, Science II 30 quai Ernest Ansermet
Geneva CH-1211
Switzerland

Dear Dr. Roux,

Thank you for submitting your manuscript entitled "Vps60 initiates formation of alternative membrane-bound ESCRT-III filaments". Your manuscript has been assessed by expert reviewers, whose comments are appended below. Although the reviewers are overall positive about this work, significant concerns unfortunately preclude publication of the current version of the manuscript in JCB.

As you will see, the reviewers show interest in your premise and appraise the quality of your study. However, all of them raise overlapping concerns about the lacking *in vivo* evidence and relevance of Vps60-based filaments, thus, it would be required that you at least investigate the functional role of Vps60 in MVB formation in yeast and perform rescue assays using expression of Vps60 mutant proteins in cells, as suggested by reviewer #2. In addition, more evidence to support that Vps60 aggregates can mediate membrane remodeling *in vitro* would be needed. All the technical points raised by the reviewers would need to also be addressed.

Please let us know if you are able to address the major issues outlined above and wish to submit a revised manuscript to JCB. Note that a substantial amount of additional experimental data likely would be needed to satisfactorily address the concerns of the reviewers. We highly recommend that, before embarking yourself into revisions, you send to us a revision plan so that we can give you feedback on its suitability to satisfactorily address the reviewers' concerns -please, bear in mind that we may solicit input from reviewers.

The typical timeframe for revisions is three to four months. While most universities and institutes have reopened labs and allowed researchers to begin working at nearly pre-pandemic levels, we at JCB realize that the lingering effects of the COVID-19 pandemic may still be impacting some aspects of your work, including the acquisition of equipment and reagents. Therefore, if you anticipate any difficulties in meeting this aforementioned revision time limit, please contact us and we can work with you to find an appropriate time frame for resubmission. Please note that papers are generally considered through only one revision cycle, so any revised manuscript will likely be either accepted or rejected.

If you choose to revise and resubmit your manuscript, please also attend to the following editorial points. Please direct any editorial questions to the journal office.

GENERAL GUIDELINES:

Text limits: Character count is < 40,000, not including spaces. Count includes title page, abstract, introduction, results, discussion, and acknowledgments. Count does not include materials and methods, figure legends, references, tables, or supplemental legends.

Figures: Your manuscript may have up to 10 main text figures. To avoid delays in production, figures must be prepared according to the policies outlined in our Instructions to Authors, under Data Presentation, <https://jcb.rupress.org/site/misc/ifora.xhtml>. All figures in accepted manuscripts will be screened prior to publication.

*****IMPORTANT:** It is JCB policy that if requested, original data images must be made available. Failure to provide original images upon request will result in unavoidable delays in publication. Please ensure that you have access to all original microscopy and blot data images before submitting your revision. ***

Supplemental information: There are strict limits on the allowable amount of supplemental data. Your manuscript may have up to 5 supplemental figures. Up to 10 supplemental videos or flash animations are allowed. A summary of all supplemental material should appear at the end of the Materials and methods section.

Please note that JCB now requires authors to submit Source Data used to generate figures containing gels and Western blots with all revised manuscripts. This Source Data consists of fully uncropped and unprocessed images for each gel/blot displayed in the main and supplemental figures. Since your paper includes cropped gel and/or blot images, please be sure to provide one Source Data file for each figure that contains gels and/or blots along with your revised manuscript files. File names for Source

Data figures should be alphanumeric without any spaces or special characters (i.e., SourceDataF#, where F# refers to the associated main figure number or SourceDataFS# for those associated with Supplementary figures). The lanes of the gels/blots should be labeled as they are in the associated figure, the place where cropping was applied should be marked (with a box), and molecular weight/size standards should be labeled wherever possible.

If you choose to resubmit, please include a cover letter addressing the reviewers' comments point by point. Please also highlight all changes in the text of the manuscript.

Regardless of how you choose to proceed, we hope that the comments below will prove constructive as your work progresses. We would be happy to discuss them further once you've had a chance to consider the points raised. You can contact the journal office with any questions, cellbio@rockefeller.edu.

Thank you for thinking of JCB as an appropriate place to publish your work.

Sincerely,

Pier Paolo Di Fiore, MD, PhD
Editor
The Journal of Cell Biology

Lucia Morgado-Palacin, PhD
Scientific Editor
Journal of Cell Biology

Reviewer #1 (Comments to the Authors (Required)):

The manuscript 'Vps60 initiates formation of alternative membrane-bound ESCRT-III filaments' provides state-of-the-art in vitro reconstitution experiments to examine the assembly of different ESCRT-III filaments.

These experiments have been pioneered by Roux and his team. In this paper they set out to compare the assembly and the properties of Snf7 based ESCRT-III filaments with Vps60 based filaments. They used a variety of different assays and report interesting similarities and difference during in vitro assembly and disassembly processes. On flat membranes Vps60 can form spontaneous ring-like filaments, whereas Snf7 rings continue to grow into large spiral like structures. The Vps60 polymers from independent of Snf7 (and vice versa) and the two filaments don't seem to interact with one another. The formation of Snf7 polymers can be stimulated by nucleators such as Vps20 or Bro1. Vps60 polymer formation is not stimulated by these factors. Vps60 polymers can recruit Vps24-Vps2-Did2-Ist1. This reflects (with some interesting differences) the assembly reaction early characterized for Snf7 based ESCRT-III polymers. Upon addition Vps4, Snf7 based ESCRT-III polymers are completely disassembled while Vps60-based polymers are not, because Vps60 appears to be refractory to Vps4 mediated disassembly (under all conditions investigated here.). I only have minor comments regarding the in vitro experiments, and I'm sure that all of them can be addressed easily (please see below).

Major concern: The conclusion drawn here are based on in vitro experiments, which have been executed with great care. At the moment, it remains unclear how these findings translate into a cellular context.

Fig.8 provides beautiful 3D tomography studies from MVBs in WT and Vps60 deficient budding yeast cells. Yet, these findings are not new and have been reported earlier ([doi/10.1091/mbc.e09-09-0776](https://doi.org/10.1091/mbc.e09-09-0776)). In the original publication, phenotypes appeared a bit more severe and it was concluded that Vps60 deficient cells form 'vesicular tubular endosomes' (VTEs) with a clear defect in MVB sorting. The same paper also reported that Vps60 was not essential for the recruitment of Did2 to endosomes. Rather, deletion of Vps60 enhanced localization of GFP-Did2 at endosomes. Along the same lines, a different paper showed that Vps60 displays a strong MVB sorting defect and that Vps60 was also not essential in yeast cells for the recruitment of Ist1 and ([10.1091/mbc.e07-07-0694](https://doi.org/10.1091/mbc.e07-07-0694)). There are certainly ways to align the results of the in vivo experiments with the results of the in vitro experiments presented here.

However, as long as we lack evidence (a) that Vps60 indeed forms a polymer on membrane in vivo and (b) that this Vps60 polymer is required to recruit Vps24-Vps2-Did2-Ist1 and Vps4, I would recommend to tone down the conclusions of the in vitro experiments with respect to cellular context. Rather, I would recommend to highlight the power of the in vitro approaches. The

title of paper and the abstract and conclusion throughout the paper should reflect these changes.

Minor comments:

1. What are the Vps60 bubbles / blebs in Fig 1 G?
2. The authors report that they did not observe that Vps2, Vps24 and Did2 bind specifically to Vps60 puncta only, but find that their binding to the membrane is specifically enhanced in the presence of Vps60 puncta. As Vps60-filaments do not form patches, it may be that Vps60 serves as a nucleation template for ESCRT-III polymers which can then diffuse along membranes. Could this dissociation of Vps2-Vp24, Did2 be responsible for the apparent failure of Vps4 to disassembly Vps60?
3. I believe that in Fig. 4B the color code or the labeling was switched?
4. In Fig.4D, there seem to be green dots in areas without membranes and the frequencies appear similar to areas with membranes? Please clarify maybe I'm misinterpreting the result here.
5. If Vps60 assembles indeed into Vps4 resistant rings, it seems that it would persist on membranes once it is recruited and could not be reused for other cellular processes. How that translate into an in vivo setting? The author speculate that another AAA+ protein might be responsible to remove Vps60. This argument is well taken. Could the authors speculate how this idea is compatible with the in vivo finding that Vps60-GFP (whose function is probably compromised), only partially disrupted Vps24, and Snf7, recycling off endosomes, and required Vps4 to be recruited to class-E compartments?
6. The authors conclusion that 'Vps60- and Snf7-based polymers assemble and undergo disassembly independently of each other' is a bit confusing. I think the paper shows clearly that Vps60 polymers don't disassemble.
7. Does the model in Fig.8 summarize the findings? Does it take into consideration that the Vps24-Vps2-Did2-Ist1 modul detaches from Vps60 polymers?

Reviewer #2 (Comments to the Authors (Required)):

In this manuscript, Pfitzner and colleagues demonstrate that the ESCRT-III protein Vps60 can form membrane-bound polymers, which can recruit other ESCRT-III subunits and disassemble in a Vps4-dependent manner and propose that these assemblies are distinct from the canonical Snf7-based polymers.

ESCRT-III filaments mediate membrane remodeling and scission events needed in a variety of cellular processes spanning from cytokinetic abscission to homeostasis of intracellular organelles like multivesicular bodies (MBVs) or the nucleus. The idea that different types of ESCRT-III filaments could perform alternative functions, providing an additional mechanism to regulate the functional specificity of the ESCRT pathway, is conceptually very novel and exciting. However, I find that the evidence provided to demonstrate the physiological relevance of the Vps60-based filaments is not strong enough, making this study too premature for publication.

My main concerns are:

1. One of the limitations of the study is that, although the authors demonstrate that Vps60 can bind membranes and spontaneously polymerize very efficiently, this doesn't demonstrate that Vps60 aggregates can mediate membrane remodeling. One way to do this would be to show inward protrusions and/or membrane constriction in negative-stain electron micrographs of LUVs incubated with Vps60-Vps2-Vps24-Did2, similarly to what has been previously done with Snf7-based filaments (Cashikar et al, 2014 & Pfitzner et al, 2020)
2. To demonstrate the physiological relevance of the newly observed Vps60 filaments, the authors look at MVB formation in yeast and show that Vps60 deletion reduces ILV formation. First, this is not a novel observation; Vps60 Δ mutants have previously been classified as having weak protein sorting phenotypes (Rue et al., 2007) and have been shown to have MVB morphology defects (Nickerson et al, 2010). Secondly, this doesn't demonstrate that the defects observed are due to inadequate Vps60 polymerization (an alternative explanation could be that Vps60, through its interaction with Vta1, is needed for efficient Vps4 activity on Snf7-based filaments). To do so, expression of Vps60 mutant proteins (e.g., unable to bind membrane, ESCRT-III or Vta1) in cells could be used to rescue the observed phenotypes.

Minor text change needed: labeling of Fig1 is not correct - There are more panels (A-J) than the described in the figure legend (A-D) and there is a mismatch with the mention to the specific panels in the main text (for example Fig. 1E should be Fig. 1G)

Reviewer #3 (Comments to the Authors (Required)):

This manuscript deals with the biophysical properties of the ESCRT-III subunit Vps60 to form filaments, associate with membranes, and recruit downstream ESCRT-III components to membranes. Using in vitro assays with purified proteins and

either GUVs or supported lipid bilayers, the authors found some similarities between Vps60 and Snf7, the main components of the ESCRT-III spiral filaments that mediate formation of intraluminal vesicles (ILVs) in multivesicular endosomes or MBVs. Based on the similarities and differences in the behaviour of Vps60 and Snf7 in these in vitro assays, the authors propose that Vps60 and its associated ESCRT assembly may have gained specific cellular functions.

Vps60 is a poorly characterized ESCRT subunit, usually considered as an "accessory" ESCRT-III subunit in yeast, not essential for MVB sorting. Thus, this manuscript sheds light on a very important topic in membrane trafficking by analysing biophysical properties of Vps60 related to its ability to bind membranes, nucleate and recruit ESCRT assemblies. The in vitro experiments are of very good quality although a number of comments and suggestions are included below for better appreciation and interpretation of results. However, the only in vivo evidence of Vps60 function in yeast presented in the manuscript (analysis of ILV density in wild type and vps60 mutant yeast MVBs) does not support a specific function of Vps60 independent of Snf7 and instead, shows a minor role of Vps60 in ILV formation.

Comments

1) It would be important to indicate which fluorescent probe was used to label each protein in each experiment. Their ability to bind proteins is highly dependent on the pH of the solution. For example, in Figure 1 pH variations lead to differential association of ESCRT-III proteins to membranes, it would be important to discern whether the dye itself could be introducing artifacts.

2) A more detailed legend should explain what exactly the figure is depicting. Are the individual lines separate experiments? What is the kymogram on the top an example of a representative experiment? What is the kymogram in panel C showing? Why does the membrane look so different from those in panel A and B? The membrane in D looks also different from those in panels A and B. What is the green probe used to visualize VPS60 in panel B? The authors should also label which rings in panel H are curled or stacked, as described in the main text.

3) Line 107 "Likewise, Vps60 is strongly recruited along the GUV's flat membrane, whereas only minor binding is observed along the highly curved nanotube (Fig. 1E, S1E)". It would be useful to include an estimation of the membrane curvature of the GUV since no scale bar is included to appreciate its size. I can't see the "minor binding" to nanotubule in Fig 1G or in Fig S1E. The authors may want to choose a better image, label the image, and quantify how much protein signal is associated with the nanotubule.

4) Line 146: "In contrast to the estimated nucleation rate, which was likely underestimated due to overlapping Vps60 puncta, images at concentrations between 150 nM and 250 nM do not suggest a saturation of Vps60 binding." The legend in Figure 2 does not explain clearly what the panels show. For example, is the graph in panel B showing the number of VPS60 puncta as a proxy for nucleation events or total Vps60 fluorescence? How is this graph demonstrating lack of binding saturation as stated in the sentence in line 146? The curve seems to be plateauing between 150 and 250nm which would indicate saturation.

5) Line 156. "Observed filament structures of Vps60 (Fig. 1C) suggested that filament breaking might occur less frequently" Fig 1H, not C shows VPS60 filaments. The filaments depicted here are mostly rings with a few examples of 2-4 stacked rings. How do these images support the notion that these filaments break less easily than those of Snf7?

6) Line 200: "To test our hypothesis, we studied ESCRT-III subunit binding to SLBs pre-incubated with Vps60. Indeed, Alexa488-Vps2 was recruited strongly to Atto565-Vps60-covered SLBs in the presence of Vps24 and Did2 (Fig. 4A-B)." This experiment lacks controls. As shown in Fig 1F, Vps2-Did2-Ist1 can be recruited to GUVs at lower pH, independently of Vps60. The quantifications displayed in Fig 4A should also include Vps2+Did2+Vps24 and Vps2+Did2+Vps24+Ist1 without Vps60 to appreciate the specific contribution of Vps60 to the recruitment of Vps2. Same observation for Vps24 in Fig 4C-D, Did2 in Fig 2E, and Ist1 in Fig 4F. Without these basic controls, it is almost impossible to assess the validity of the claims that Vps60 polymers recruit downstream ESCRT-III subunits

7) For Fig 6, it would be useful to show quantifications of the Vps2 fluorescence co-localizing with Snf7 and with Vps60, instead of only the Vps2 fluorescence not co-localizing with Snf7

8) Tomographic analysis of ILVs: Based on the images shown in Fig 7, it looks like besides the density of ILVs, there could be differences in number, diameter of MVBs, diameter of ILVs between wild type and the vps60 mutant. How many cells and how many MVB tomograms were used for this analysis? The relative weak phenotype of the vps60 mutant ILVs and the fact that Vps60 requires Snf7 for endosomal membrane recruitment, does not seem to support a Vps60 function independent from Snf7

9) No statistical analyses are included in most of the quantifications.

Minor comments:

Line 104 "... (See Methods and Fig 1E)" It is actually panel G, not E the one that shows how the nanotubules are generated.

Line 106 "As previously reported, Snf7 bound exclusively to flat membrane (Fig. 1E, S1C)" I believe this is described in Fig 1G. It

is therefore unclear what Figure 1E and F are showing. I assume they are described in the previous paragraph under Fig1D?

Lines 115-116: "Indeed, Vps60 formed ring-shaped filaments with an average diameter of $18.9\text{\AA} \pm 3.4$ nm (Fig. 1F-H)". Once again, the panels are mis-called in the text. It should be Fig 1I-J, not Fig 1F-H). The panels in Fig 1 are mis-called in the following paragraphs as well.

Line 144, define SLB

Line 248: "triggred" typo

Line 253: "Vp60" typo

Line 338: "subunit" should be "subunits"

Line 346: "filament" should be "filaments"

In the methods section "Supported membrane bilayer assay": "Supported membrane bilayer assay was performed as described in _____." Missing citation

Reviewer #1 (Comments to the Authors (Required)):

The manuscript 'Vps60 initiates formation of alternative membrane-bound ESCRT-III filaments' provides state-of-the-art in vitro reconstitution experiments to examine the assembly of different ESCRT-III filaments.

These experiments have been pioneered by Roux and his team. In this paper they set out to compare the assembly and the properties of Snf7 based ESCRT-III filaments with Vps60 based filaments. They used a variety of different assays and report interesting similarities and difference during in vitro assembly and disassembly processes. On flat membranes Vps60 can form spontaneous ring-like filaments, whereas Snf7 rings continue to grow into large spiral like structures. The Vps60 polymers from independent of Snf7 (and vice versa) and the two filaments don't seem to interact with one another. The formation of Snf7 polymers can be stimulated by nucleators such as Vps20 or Bro1. Vps60 polymer formation is not stimulated by these factors. Vps60 polymers can recruit Vps24-Vps2-Did2-Ist1. This reflects (with some interesting differences) the assembly reaction early characterized for Snf7 based ESCRT-III polymers. Upon addition Vps4, Snf7 based ESCRT-III polymers are completely disassembled while Vps60-based polymers are not, because Vps60 appears to be refractory to Vps4 mediated disassembly (under all conditions investigated here.). I only have minor comments regarding the in vitro experiments, and I'm sure that all of them can be addressed easily (please see below).

We thank the reviewer for the positive evaluation of our study, and we hope that we have answered all remaining concerns discussed below.

Major concern: The conclusion drawn here are based on in vitro experiments, which have been executed with great care. At the moment, it remains unclear how these findings translate into a cellular context.

This is a fair point, and we have largely extended the in vivo data to provide more insights into the cellular context. In particular we have added the following data:

1-Density gradient experiments that show that Vps60 lives as a polymer in the yeast, as previously shown for Snf7. However, its behaviour upon inhibition of Vps4 function is different than Snf7, as it goes back to monomeric form, while Snf7 sees further increase of its polymeric form.

2-We compared the recruitment, the localisation and the dynamics of Chmp5 - the mammalian homolog of Vps60 - and Chmp4 - the mammalian homolog of Snf7 - in several cellular function of human fibroblasts. We can show that chmp5 is recruited during cytokinesis and endosomal functions, but its localisation is segregated from chmp4, and its recruitment dynamics much longer, all this being consistent with our in vitro data. Also, we can show that Chmp5 is not recruited during Nuclear Envelope Reformation, while Chmp4 is.

All these data support that the in vitro findings we proposed in the initial version of our manuscript transcribe well in the cellular context of both yeast and mammalian cells.

Fig.8 provides beautiful 3D tomography studies from MVBs in WT and Vps60 deficient budding yeast cells. Yet, these findings are not new and have been reported earlier ([doi/10.1091/mbc.e09-09-0776](https://doi.org/10.1091/mbc.e09-09-0776)). In the original publication, phenotypes appeared a bit more severe and it was concluded that Vps60 deficient cells form 'vesicular tubular endosomes' (VTEs) with a clear defect in MVB sorting. The same paper also reported that Vps60 was not essential for the recruitment of Did2 to endosomes. Rather, deletion of Vps60 enhanced localization of GFP-Did2 at endosomes. Along the

same lines, a different paper showed that Vps60 displays a strong MVB sorting defect and that Vps60 was also not essential in yeast cells for the recruitment of Ist1 and (10.1091/mbc.e07-07-0694). There are certainly ways to align the results of the in vivo experiments with the results of the in vitro experiments presented here.

The results discussed above are in full agreement with our in vitro findings that Vps60 and Snf7 have similar biochemical properties, but segregated assemblies and functions. That Did2 and Ist1 recruitment are not strongly affected by Vps60 deletion is consistent with the fact that they can be recruited by both Snf7 and Vps60. In this case, deletion of Vps60 could be compensated by Snf7-mediated recruitment. Also, the defect in MVB formation that we show in Fig. 8, and the stronger phenotype described in the paper cited by this reviewer, are suggesting - not showing - that Vps60 and Snf7 could participate in distinct pathways to generate ILVs.

These points have been added to the discussion and the papers above are cited, lines 339-342 and L415-428 of the revised manuscript.

However, as long as we lack evidence (a) that Vps60 indeed forms a polymer on membrane in vivo and (b) that this Vps60 polymer is required to recruit Vps24-Vps2-Did2-Ist1 and Vps4, I would recommend to tone down the conclusions of the in vitro experiments with respect to cellular context. Rather, I would recommend to highlight the power of the in vitro approaches. The title of paper and the abstract and conclusion throughout the paper should reflect these changes.

This comment of the reviewer triggered us into performing more in vivo experiments and now show that Vps60 is present as membrane-bound polymers of similar size than Snf7-polymers in cells (Fig.7). We also find that these Vps60-polymers don't form higher molecular polymers when Vps4 function is disrupted, which, in alignment with our in vitro data, indicates that their recycling is not dependent on Vps4. As discussed above, the papers cited by this reviewer show that recruitment of the complete sequence of proteins in yeast is not strongly affected by the deletion of Vps60. It may thus be difficult, in the context of redundant recruitment of these proteins by Snf7, to clearly show that Vps60 can recruit subunits downstream in the sequence. This should be done in the Snf7 deletion mutant, which has a much more severe phenotype, precluding the possibility to investigate the role of Vps60 in this mutant.

Minor comments:

1. What are the Vps60 bubbles / blebs in Fig 1 G?

These are Vps60 polymers formed in solution under specific experimental conditions. The experimental conditions were chosen to keep the same concentration of all subunits. We now mentioned this fact in the text (Lines 104-6) and highlighted the lack of membrane in the polymers in the supplementary figure S1C.

“Vps60 puncta observed outside the nanotube or GUV outlines most likely correspond to protein polymers as they do not contain membrane (Fig. S1C).”

2. The authors report that they did not observe that Vps2, Vps24 and Did2 bind specifically to Vps60 puncta only, but find that their binding to the membrane is specifically enhanced in the presence of Vps60 puncta. As Vps60-filaments do not form patches, it may be that Vps60 serves as a nucleation

template for ESCRT-III polymers which can then diffuse along membranes. Could this dissociation of Vps2-Vp24, Did2 be responsible for the apparent failure of Vps4 to disassembly Vps60?

This is indeed a very thoughtful interpretation of our data, and we now discuss this possibility in the text (line 198-202). The new negative stain EM data could also be interpreted further in this sense: we observe multistranded Vps60-based spirals (Fig. 4E) together with Vps2-Vps24-Did2, which argues against the detachment of subunits from the Vps60 initial ring. At the meantime, in the same experiments, Vps2-Vps24 polymers formed in bulk are decorated at one tip with opened rings most probably of Vps60 (fig4F). This could be an intermediate of the "detached" form proposed by this reviewer. It is however interesting to notice that the two forms in our EM grids are distinguished by the presence or absence of the membrane. Thus, it could be that the membrane regulates the association/dissociation of ESCRT-III filaments made of different subunits.

3. I believe that in Fig. 4B the color code or the labeling was switched?

Thank you we corrected this.

4. In Fig.4D, there seem to be green dots in areas without membranes and the frequencies appear similar to areas with membranes? Please clarify maybe I'm misinterpreting the result here. the (labelled) proteins are absorbed by the glass slides outside of the membrane areas. However binding to the membrane can be specifically seen upon fluorescence increase on the membrane as otherwise the fluorescence is excluded from the membrane areas as can be seen in Fig. 4 D Vps24 or Vps24-Did2 controls.

5. If Vps60 assembles indeed into Vps4 resistant rings, it seems that it would persist on membranes once it is recruited and could not be reused for other cellular processes. How that translate into an in vivo setting? The author speculate that another AAA+ protein might be responsible to remove Vps60. This argument is well taken. Could the authors speculate how this idea is compatible with the in vivo finding that Vps60-GFP (whose function is probably compromised), only partially disrupted Vps24, and Snf7, recycling off endosomes, and required Vps4 to be recruited to class-E compartments?

We added data to show that Vps60 in vivo does not accumulate in higher molecular weight polymers in respond to lack of Vps4 activity (Fig. 7) which supports the notion, based on our in vitro data, that Vps4 is not the primary recycling ATPase of Vps60. In contrast, Vps60 returns to monomeric form in these conditions, which strongly suggests that another factor participates in its depolymerization. We also tested in vitro if Vps60 could be integrated into Snf7-based ESCRT-III polymers upon Vps4 activity (Fig S3C-D) which was not the case in presence of all other ESCRT-III subunits. We cannot exclude that addition factors in cells would allow integration of Vps60 in the polymers upon Vps4 action, however this still leaves the question of the Vps60 recycling opened. It presumably requires an additional AAATPase, which could be Spastin in multicellular organisms. But Spastin being absent from the Yeast genome, it is difficult to conclude generally. Another possibility is that the ATPase activity could be directly associated with the Vps60 polymer, as it is the case for Bacterial (Vipp1) and Chloroplastic ESCRT-III (PspA).

6. The authors conclusion that 'Vps60- and Snf7-based polymers assemble and undergo disassembly independently of each other' is a bit confusing. I think the paper shows clearly that Vps60 polymers don't disassemble.

Vps2-Vps24-Did2-Ist1 are clearly disassembled from Vps60-based polymers, while Vps60 stays onto the membrane. So the ESCRT-III polymers generated by Vps60 clearly disassemble, only Vps60 stays on the membrane. We formulated this clearer in the manuscript in Lines 298-293.

7. Does the model in Fig.8 summarize the findings? Does it take into consideration that the Vps24-Vps2-Did2-Ist1 modul detaches from Vps60 polymers?

As discussed above, the detachment must be regulated, but the mechanism of regulation is unknown, even though membrane itself may be involved. This makes difficult to include the detachment in the model of Fig8. We nevertheless discuss this possibility shortly in the text (Line 246-250).

Reviewer #2 (Comments to the Authors (Required)):

In this manuscript, Pfitzner and colleagues demonstrate that the ESCRT-III protein Vps60 can form membrane-bound polymers, which can recruit other ESCRT-III subunits and disassemble in a Vps4-dependent manner and propose that these assemblies are distinct from the canonical Snf7-based polymers.

ESCRT-III filaments mediate membrane remodeling and scission events needed in a variety of cellular processes spanning from cytokinetic abscission to homeostasis of intracellular organelles like multivesicular bodies (MBVs) or the nucleus. The idea that different types of ESCRT-III filaments could perform alternative functions, providing an additional mechanism to regulate the functional specificity of the ESCRT pathway, is conceptually very novel and exciting. However, I find that the evidence provided to demonstrate the physiological relevance of the Vps60-based filaments is not strong enough, making this study too premature for publication.

We thank this reviewer for the overall positive assessment of our manuscript, we have substantially strengthen the in vivo study, giving support to the physiological relevance of our in vitro data.

My main concerns are:

1. One of the limitations of the study is that, although the authors demonstrate that Vps60 can bind membranes and spontaneously polymerize very efficiently, this doesn't demonstrate that Vps60 aggregates can mediate membrane remodeling. One way to do this would be to show inward protrusions and/or membrane constriction in negative-stain electron micrographs of LUVs incubated with Vps60-Vps2-Vps24-Did2, similarly to what has been previously done with Snf7-based filaments (Cashikar et al, 2014 & Pfitzner et al, 2020)

We observed some membrane deformation and membrane bound double-stranded spirals with the Vps60-based polymers. The images are included in S3E. The deformations observed were less specific for the Vps60-based filaments than for the Snf7-based for which we were able to map them. It is important to consider that most of the membrane remodelling properties of ESCRT-III filaments are observed during Vps4-induced turnover. As discussed with reviewer 1, it seems that the exchange factor of Vps60 polymers is missing in our experiments, as it is not Vps4. We are thus lacking an important factor for characterizing fully the membrane remodelling properties. In order to establish fully the membrane remodelling capacity of Vps60 polymers, we would need to identify, characterize and reconstitute membrane-remodelling by Vps60 with the missing factor, a task beyond the scope of this paper.

2. To demonstrate the physiological relevance of the newly observed Vps60 filaments, the authors look at MVB formation in yeast and show that Vps60 deletion reduces ILV formation. First, this is not a novel observation; Vps60Δ mutants have previously been classified as having weak protein sorting phenotypes (Rue et al., 2007) and have been shown to have MVB morphology defects (Nickerson et al, 2010). Secondly, this doesn't demonstrate that the defects observed are due to inadequate Vps60 polymerization (an alternative explanation could be that Vps60, through its interaction with Vta1, is needed for efficient Vps4 activity on Snf7-based filaments). To do so, expression of Vps60 mutant proteins (e.g., unable to bind membrane, ESCRT-III or Vta1) in cells could be used to rescue the observed phenotypes.

This is a fair comment from this review. As Vps60 mutants deficient in either polymerizing or binding Vps2, Vps24 are not known, we have however taken another approach to show the physiological relevance of our in vitro findings. Even for Snf7 which is the one of the most heavily studied ESCRT-III protein mutant that only disrupt polymerization or Vps2-Vps24 interaction are not described most likely as both are intertwined, thus finding such mutants for Vps60 seems unlikely. In order to address the physiological relevance, we added more in vivo data looking at dynamics of recruitment of CHMP5 in several ESCRT-III functions in HeLa cells (Fig. 6) and data about Vps60 polymerization (Fig. 7) in yeast. We find clear similarities between this in vivo data and our in vitro data hinting to similar conclusions as we draw in the manuscript. We hope that this will support enough the physiological relevance of our study in the view of this review.

Minor text change needed: labeling of Fig1 is not correct - There are more panels (A-J) than the described in the figure legend (A-D) and there is a mismatch with the mention to the specific panels in the main text (for example Fig. 1E should be Fig. 1G)

Thank you we have change the legend.

Reviewer #3 (Comments to the Authors (Required)):

This manuscript deals with the biophysical properties of the ESCRT-III subunit Vps60 to form filaments, associate with membranes, and recruit downstream ESCRT-III components to membranes. Using in vitro assays with purified proteins and either GUVs or supported lipid bilayers, the authors found some similarities between Vps60 and Snf7, the main components of the ESCRT-III spiral filaments that mediate formation of intraluminal vesicles (ILVs) in multivesicular endosomes or MBVs. Based on the similarities and differences in the behaviour of Vps60 and Snf7 in these in vitro assays, the authors propose that Vps60 and its associated ESCRT assembly may have gained specific cellular functions.

Vps60 is a poorly characterized ESCRT subunit, usually considered as an "accessory" ESCRT-III subunit in yeast, not essential for MVB sorting. Thus, this manuscript sheds light on a very important topic in membrane trafficking by analysing biophysical properties of Vps60 related to its ability to bind membranes, nucleate and recruit ESCRT assemblies. The in vitro experiments are of very good quality although a number of comments and suggestions are included below for better appreciation and interpretation of results. However, the only in vivo evidence of Vps60 function in yeast presented in the manuscript (analysis of ILV density in wild type and vps60 mutant yeast MVBs) does not support a specific function of Vps60 independent of Snf7 and instead, shows a minor role of Vps60 in ILV formation.

We thank this reviewer for the positive evaluation of our manuscript.

Comments

1) It would be important to indicate which fluorescent probe was used to label each protein in each experiment. Their ability to bind proteins is highly dependent on the pH of the solution. For example, in Figure 1 pH variations lead to differential association of ESCRT-III proteins to membranes, it would be important to discern whether the dye itself could be introducing artifacts.

The experiments shown here were performed with Vps2-Alexa488, Did2-Atto565, Snf7-Alexa488 and Vps60-Alexa488. We also performed this experiment with Vps60-Atto565 and Vps2-Atto565 and behaviour was the same in the experiments. This information has been added in lines 106-108.

2) A more detailed legend should explain what exactly the figure is depicting. Are the individual lines separate experiments?

Yes.

Are the kymograms on the top an example of a representative experiment?

Yes.

What is the kymogram in panel C showing? Why does the membrane look so different from those in panel A and B?

Snf7 and Vps60 (now S1G) bind along the GUV, the kymograph depicts a cross-section of the GUV overtime. Vps2-Vps24 and Vps2-Did2-Ist1 bind along the nanotube thus the kymograph is taking along the nanotube thus the different appearance.

The membrane in D looks also different from those in panels A and B. What is the green probe used to visualize VPS60 in panel B?

Alexa488

The authors should also label which rings in panel H are curled or stacked, as described in the main text.

We included zoom-ins of the rings to better display the different architectures.

3) Line 107 "Likewise, Vps60 is strongly recruited along the GUV's flat membrane, whereas only minor binding is observed along the highly curved nanotube (Fig. 1E, S1E)". It would be useful to include an estimation of the membrane curvature of the GUV since no scale bar is included to appreciate its size.

The curvature on the GUV is neglectable compared to the nanotube curvature. The nanotubes have diameter below 200 nm, thus below the resolution limit making it difficult to measure the radius without a proper fluorescence calibration or force measurement. But the diameter of these tubes is typically between 20 and 80nm.

I can't see the "minor binding" to nanotubule in Fig 1G or in Fig S1E. The authors may want to choose a better image, label the image, and quantify how much protein signal is associated with the nanotubule.

The minor binding is often not visible, so we changed "minor binding" to "no observable binding" line 102.

4) Line 146: "In contrast to the estimated nucleation rate, which was likely underestimated due to

overlapping Vps60 puncta, images at concentrations between 150 nM and 250 nM do not suggest a saturation of Vps60 binding." The legend in Figure 2 does not explain clearly what the panels show. For example, is the graph in panel B showing the number of VPS60 puncta as a proxy for nucleation events or total Vps60 fluorescence?

We included the relevant information in the legends.

How is this graph demonstrating lack of binding saturation as stated in the sentence in line 146? The curve seems to be plateauing between 150 and 250nm which would indicate saturation.

Indeed, as the reviewer stated we observe a plateau in our quantification which would suggest a binding saturation. If we compare this with the raw images, the impression is, however, not confirmed. We thus believed that our automated quantification fails at higher concentration due to problems of distinguishing puncta and higher background fluorescence.

5) Line 156. "Observed filament structures of Vps60 (Fig. 1C) suggested that filament breaking might occur less frequently" Fig 1H, not C shows VPS60 filaments. The filaments depicted here are mostly rings with a few examples of 2-4 stacked rings. How do these images support the notion that these filaments break less easily than those of Snf7?

Upon incubation with SLB, Snf7 forms growing patches (Fig. 2C). Chiaruttini et al 2015 showed that this is due to breakage of growing spiraling filaments to give rise to two new spirals. Altogether this fuels a chain reaction which can be observed with confocal microscopy as growing patches due to reduced diffusion on SLB. In contrast, we observe for Vps60 no such growing patches but puncta and a general increase of fluorescence. We do not observe growing spiral filaments with Vps60 alone. We observe mostly rings and a few broken rings with curled in ends which could suggest that the filament were under tension before breakage. From these observations we draw the conclusion that Vps60 ring filaments break less often than Snf7 filaments. We use the same membrane composition and protein concentration in the experiments. We also added some data that show spiral formation of Vps60 filaments upon recruitment of downstream subunits, which demonstrates the potential of Vps60 filament to undergo transition from rings to spirals, however, in our experimental conditions they don't in contrast to Snf7 filaments which readily undergo the transition to spirals alone.

6) Line 200: "To test our hypothesis, we studied ESCRT-III subunit binding to SLBs pre-incubated with Vps60. Indeed, Alexa488-Vps2 was recruited strongly to Atto565-Vps60-covered SLBs in the presence of Vps24 and Did2 (Fig. 4A-B)." This experiment lacks controls. As shown in Fig 1F, Vps2-Did2-Ist1 can be recruited to GUVs at lower pH, independently of Vps60. The quantifications displayed in Fig 4A should also include Vps2+Did2+Vps24 and Vps2+Did2+Vps24+Ist1 without Vps60 to appreciate the specific contribution of Vps60 to the recruitment of Vps2. Same observation for Vps24 in Fig 4C-D, Did2 in Fig 2E, and Ist1 in Fig 4F. Without these basic controls, it is almost impossible to assess the validity of the claims that Vps60 polymers recruit downstream ESCRT-III subunits.

The experiments were performed at pH 6.8, and the controls requested by this reviewers can be seen in fig S1F, G and absence of binding has been reported several times before in Chiaruttini et al., 2015, Mierzwa et al., 2017, Pfitzner et al., 2020.

7) For Fig 6, it would be useful to show quantifications of the Vps2 fluorescence co-localizing with Snf7 and with Vps60, instead of only the Vps2 fluorescence not co-localizing with Snf7

In Fig 3A, B, Vps2 fluorescence increases across the membrane in presence Vps60 but no exclusive colocalization with Vps60 puncta is observed. In Figure 6 we quantify the percent of fluorescence

not-colocalizing with Snf7 which is obtained by measuring the total fluorescence on the membrane and the fluorescence which is colocalizing with Snf7 patches. We then calculated the portion of fluorescence which is outside of Snf7 patches. The quantification for colocalization of Vps2 and Snf7 is the reverse of the shown graph. We now included this graph in the supplementary data S5H.

We reason that Vps2 fluorescence on the membrane outside of Snf7-polymers cannot be explained by Snf7 action and thus it must be due to the contribution of Vps60. As a control we use recruitment via Vps2-VPs24 which is recruited to Snf7 but we find to not be efficiently recruited by Vps60 in our SLB assay (Fig 3A,B). This is confirmed almost all of the Vps2 fluorescence colocalizing with Snf7 in this control and no change over time.

8) Tomographic analysis of ILVs: Based on the images shown in Fig 7, it looks like besides the density of ILVs, there could be differences in number, diameter of MVBs, diameter of ILVs between wild type and the vps60 mutant. How many cells and how many MVB tomograms were used for this analysis? The relative weak phenotype of the vps60 mutant ILVs and the fact that Vps60 requires Snf7 for endosomal membrane recruitment, does not seem to support a Vps60 function independent from Snf7

Concerning the lack of endosomal Vps60 recruitment in Snf7 knockout cells: loss of Snf7 massively disrupts the organisation of the endo-lysosomal system in yeast, thus a disturbance of the recruitment of Vps60 might occur during this process as secondary effect. On the other hand, we cannot exclude from our data that a coupling of the recruitment of Snf7 and Vps60 polymers exists in cells. However, we observe that both polymers behave (nucleation, recruitment, turnover) completely independently from each other on membranes in vitro and that no direct exchange from Snf7 to Vps60 via other established ESCRT-III subunits or turnover of the filaments by Vps4 occurs. We further note a similar organisation pattern in the subunit appearance in the filaments and a similar change in subunit composition of the membrane-bound polymers over time. These shared underlying principles of organization leads us to see, Snf7-based and Vps60-based sequences as two separate but analogue entities that occur independently in cells to perform membrane remodelling functions. Though we don't envision it, we cannot fully exclude a sequential coupling of both in cells however we can exclude a direct exchange from Snf7-base to Vps60 base within the same polymer or a direct subsequential coupling of both without further mediating cofactors.

9) No statistical analyses are included in most of the quantifications.

We represent all evaluated ROI in the graphs and calculate as well as display the mean and standard deviation for all experiments. The differences between average values as compared to standard deviations are sufficiently separated to conclude without having to perform statistical tests. This is common procedure in in vitro tests as data range are usually narrower than in vivo.

Minor comments:

Line 104 "... (See Methods and Fig 1E)" It is actually panel G, not E the one that shows how the nanotubules are generated.

Line 106 "As previously reported, Snf7 bound exclusively to flat membrane (Fig. 1E, S1C)" I believe this is described in Fig 1G. It is therefore unclear what Figure 1E and F are showing. I assume they are described in the previous paragraph under Fig1D?

Lines 115-116: "Indeed, Vps60 formed ring-shaped filaments with an average diameter of 18.9Å +/- 3.4 nm (Fig. 1F-H)". Once again, the panels are mis-called in the text. It should be Fig 1I-J, not Fig 1F-H). The panels in Fig 1 are mis-called in the following paragraphs as well.

Line 144, define SLB

Line 248: "triggred" typo

Line 253: "Vp60" typo

Line 338: "subunit" should be "subunits"

Line 346: "filament" should be "filaments"

In the methods section "Supported membrane bilayer assay": "Supported membrane bilayer assay was performed as described in _____. " Missing citation

All these typos were corrected.

May 2, 2023

RE: JCB Manuscript #202206028R

Prof. Aurélien Roux
University of Geneva
Biochemistry
University of Geneva, Science II
30 quai Ernest Ansermet
Geneva CH-1211
Switzerland

Dear Prof. Roux:

Thank you for submitting your revised manuscript entitled "Alternative ESCRT-III filaments initiated by Vps60". The original reviewers have now assessed your revised manuscript and, as you can see, they are overall satisfied with the revisions. However, there are some remaining reviewers' comments, including the discussion of alternative possibilities, that would need to be addressed through textual edits. We would be happy to publish your paper in JCB pending final revisions necessary to address the remaining points of the reviewers. Please, when submitting the final revision, make sure that you comply with our formatting guidelines (see details below).

To avoid unnecessary delays in the acceptance and publication of your paper, please read the following information carefully. Please go through all the formatting points paying special attention to those marked with asterisks.

A. MANUSCRIPT ORGANIZATION AND FORMATTING:

1) Text limits: Character count for Articles and Tools is < 40,000, not including spaces. Count includes title page, abstract, introduction, results, discussion, and acknowledgments. Count does not include materials and methods, figure legends, references, tables, or supplemental legends.

2) Figures limits: Articles and Tools may have up to 10 main text figures.

Please note that main text figures should be provided as individual, editable files.

3) Figure formatting:

******* Molecular weight or nucleic acid size markers must be included on all gel electrophoresis. Please, add MW size markers to Fig 6F.

******* Scale bars must be present on all microscopy images, including inset magnifications. Please, add scale bars to Fig 1A, 8A (inset magnifications), S1B-E, S2F, and S3C-D.

******* Also, please avoid pairing red and green for images and graphs to ensure legibility for color-blind readers. As red and green are paired for images in Fig 2F, 2H-I, 6B, 6D, S1D, and S1F, please ensure that the particular red and green hues used in micrographs are distinctive with any of the colorblind types (you can use color blindness simulators available online). If not, please modify colors accordingly or provide separate images of the individual channels.

4) Statistical analysis:

******* Error bars on graphic representations of numerical data must be clearly described in the figure legend.

******* The number of independent data points (n) represented in a graph must be indicated in the legend. Please, indicate whether N refers to technical or biological replicates (i.e. number of analyzed cells, samples or animals, number of independent experiments). Please, indicate 'n' in Fig 6E, 7F, S2B, and S4B.

If independent experiments with multiple biological replicates have been performed, we recommend using distribution-reproducibility SuperPlots (please, see Lord et al., JCB 2020) to better display the distribution of the entire dataset, and report

statistics (such as means, error bars, and P values) that address the reproducibility of the findings.

Statistical methods should be explained in full in the materials and methods in a separate section.

For figures presenting pooled data the statistical measure should be defined in the figure legends.

Please also be sure to indicate the statistical tests used in each of your experiments (both in the figure legend itself and in a separate methods section) as well as the parameters of the test (for example, if you ran a t-test, please indicate if it was one- or two-sided, etc.).

*** As you used parametric tests in your study (i.e. t-tests), you should have first determined whether the data was normally distributed before selecting that test. In the stats section of the methods, please indicate how you tested for normality. If you did not test for normality, you must state something to the effect that "Data distribution was assumed to be normal but this was not formally tested."

5) Abstract and title:

The abstract should be no longer than 160 words and should communicate the significance of the paper for a general audience.

*** The title should be less than 100 characters including spaces. Make the title concise but accessible to a general readership. We suggest changing the title to the active voice: "Vps60 initiates alternative ESCRT-III filaments."

6) Materials and methods:

Should be comprehensive and not simply reference a previous publication for details on how an experiment was performed. The text should not refer to methods "...as previously described."

Also, the materials and methods should be included in the main manuscript text and not in the supplementary materials.

7) For all cell lines, vectors, constructs/cDNAs, etc. -all genetic material: please include database/vendor ID (e.g., Addgene, ATCC, etc.) or if unavailable, please briefly describe their basic genetic features, even if described in other published work or gifted to you by other investigators (and provide references where appropriate).

Please be sure to provide the sequences for all of your oligos: primers, si/shRNA, RNAi, gRNAs, etc. in the materials and methods.

*** You must also indicate in the methods the source, species, and catalog numbers/vendor identifiers (where appropriate) for all your antibodies, including secondary, and the system used to collect the signal from the antibodies. If antibodies are not commercial, please add a reference citation if possible. Please, indicate the system/machine used to acquire the signal of your antibodies.

8) Microscope image acquisition:

The following information must be provided about the acquisition and processing of images:

- a. Make and model of microscope
- b. Type, magnification, and numerical aperture of the objective lenses
- c. Temperature
- d. imaging medium
- e. Fluorochromes
- f. Camera make and model
- g. Acquisition software
- h. Any software used for image processing subsequent to data acquisition. Please include details and types of operations involved (e.g., type of deconvolution, 3D reconstitutions, surface or volume rendering, gamma adjustments, etc.).

10) Supplemental materials:

There are strict limits on the allowable amount of supplemental data. Articles and Tools may have up to 5 supplemental figures. There is no limit for supplemental tables.

Please note that supplemental figures and tables should be provided as individual, editable files.

*** A summary of all supplemental material should appear at the end of the Materials and Methods section (please see any recent JCB paper for an example of this summary).

11) Video legends:

Video legends should describe what is being shown, the cell type or tissue being viewed (including relevant cell treatments, concentration and duration, or transfection), the imaging method (e.g., time-lapse epifluorescence microscopy), what each color represents, how often frames were collected, the frames/second display rate, and the number of any figure that has related video stills or images.

12) eTOC summary:

*** A ~40-50 word summary that describes the context and significance of the findings for a general readership should be included on the title page. The statement should be written in the present tense and refer to the work in the third person. It should begin with "First author name(s) et al..." to match our preferred style.

13) Conflict of interest statement:

*** JCB requires inclusion of a statement in the acknowledgements regarding competing financial interests. If no competing financial interests exist, please include the following statement: "The authors declare no competing financial interests."

14) Author contribution:

A separate author contribution section is required following the Acknowledgments in all research manuscripts.

*** All authors should be mentioned and designated by their first and middle initials and full surnames and the CRediT nomenclature is encouraged (<https://casrai.org/credit/>).

15) ORCID IDs: ORCID IDs are unique identifiers allowing researchers to create a record of their various scholarly contributions in a single place. At resubmission of your final files, please consider providing an ORCID ID for as many contributing authors as possible.

16) Materials and data sharing:

All animal and human studies must be conducted in compliance with relevant local guidelines, such as the US Department of Health and Human Services Guide for the Care and Use of Laboratory Animals or MRC guidelines, and must be approved by the authors' Institutional Review Board(s). A statement to this effect with the name of the approving IRB(s) must be included in the Materials and Methods section.

*** Journal of Cell Biology now requires a data availability statement for all research article submissions. These statements will be published in the article directly above the Acknowledgments. The statement should address all data underlying the research presented in the manuscript. Please visit the JCB instructions for authors for guidelines and examples of statements at (<https://rupress.org/jcb/pages/editorial-policies#data-availability-statement>).

*** We discourage making data available upon request and prefer that it be publicly available. If there are privacy concerns that prevent the data from being publicly disclosed, we request that you provide a description of the reasons for this, along with contact information and conditions for re-use. However, if there are no privacy concerns, we request that you upload your data to Dryad, which is now available on the JCB submission portal and free for our authors to use.

All datasets included in the manuscript must be available from the date of online publication, and the source code for all custom computational methods, apart from commercial software programs, must be made available either in a publicly available database or as supplemental materials hosted on the journal website. Numerous resources exist for data storage and sharing (see Data Deposition: <https://rupress.org/jcb/pages/data-deposition>), and you should choose the most appropriate venue based on your data type and/or community standard. If no appropriate specific database exists, please deposit your data to an appropriate publicly available database.

17) Please note that JCB now requires authors to submit Source Data used to generate figures containing gels and Western blots with all revised manuscripts. This Source Data consists of fully uncropped and unprocessed images for each gel/blot displayed in the main and supplemental figures. The Source Data files will be directly linked to specific figures in the published article.

As your paper includes cropped gel and/or blot images, please be sure to provide one Source Data file for each figure that contains gels and/or blots along with your revised manuscript files. File names for Source Data figures should be alphanumeric without any spaces or special characters (i.e., SourceDataF#, where F# refers to the associated main figure number or SourceDataFS# for those associated with Supplementary figures). The lanes of the gels/blots should be labeled as they are in the associated figure, the place where cropping was applied should be marked (with a box), and molecular weight/size standards should be labeled wherever possible.

B. FINAL FILES:

Thank you for this interesting contribution, we look forward to publishing your paper in Journal of Cell Biology.

Sincerely,

Pier Paolo Di Fiore, MD, PhD
Editor
The Journal of Cell Biology

Lucia Morgado-Palacin, PhD
Scientific Editor
Journal of Cell Biology

Reviewer #1 (Comments to the Authors (Required)):

The authors have submitted a marvelous revision of their original manuscript. My original concerns have been addressed. The new experiments in yeast and human cells are in line with their *in vitro* reconstitution experiments and strongly support the conclusion that Vps60 (Chmp5) and Snf7 (Chmp4) form alternative ESCRT-III filaments.

I spotted a typo in the abstract: line 22 '...may have enabled...'

Reviewer #2 (Comments to the Authors (Required)):

In their revised article, the authors have produced new data that strengthens further the *in vitro* part of the study (new Fig4 and Sup Fig3 E-G) and show that the dynamics of assembly-disassembly for Vps60 and Snf7 are different in yeast cells (new Fig 6F-G), providing some initial cellular context to their findings. They have also investigated CHMP5 localization in mammalian cells during different ESCRT-related processes (Figs 7-8) and show that CHMP5 might not be needed for some of the pathway's cellular functions (NE reformation).

Comments:

1) The *in vitro* data are very strong but unequivocal demonstration of a specific function of Vps60-based filaments independently of Snf7 *in vivo* is still missing. Thus, the words "alternative" and "functionally" (distinct polymers, line 30) should be removed from the title and abstract, respectively.

2) Can you please clarify why is Vps60 co-sedimenting at the molecular weight corresponding to the monomeric complex in the absence of functional Vps4? Wouldn't you expect to see that higher molecular weight polymers are not present (as opposed to what's observed with Snf7) but still see a polymer ~440 KD like the one in WT conditions (I'm assuming that the blots labelled as VPS4 in Fig 6F are WT cells)? How does Vps60 look in the cytoplasmic fraction?

3) Whilst I appreciate that finding a point mutant that specifically abrogates ESCRT-III binding might not be possible, there are mutants already described that abolish the interaction between CHMP5/Vps60 and VTA1/LIP5 (doi: 10.1074/jbc.M112.417899) and finding a mutant that doesn't bind membranes should be possible based on the studies done with CHMP3 (DOI: 10.1016/j.devcel.2006.03.013). These mutants would rule out the possibility that Vps60, through its interaction with Vta1, is needed for efficient Vps4 activity on Snf7-based filaments or show that Vps60 membrane binding, independently of a possible interaction with Snf7, is needed for its function. These concerns have not been addressed.

4) Could the fact that the dynamic of CHMP5 recruitment is delayed when compared to that of CHMP4B mean that CHMP5 is simply downstream of CHMP4B rather than segregated from CHMP4B? For example, as shown in Fig. 8C, CHMP5 is recruited to the intercellular bridge later than CHMP4B (CHMP4B appears in frame 32 min, whilst CHMP5 in 56 min) and this could similarly explain the delayed kinetics of CHMP5 recruitment to endosomes. Experiments with live cells co-expressing CHMP4B and CHMP5 (like the ones used in Fig 8 B&C) would have been better controlled in order to compare recruitment dynamics of these two proteins.

5) How many cells have the authors analyzed in the experiments shown in Fig 7 and 8? Sample size should be included in the figure legend. Also, is the CHMP4C stained cell shown in Fig 8A (bottom panel) the same as the one shown in the +LLoMe panel in Fig 7C? If so, this should be indicated in the figure legend. The cell shown in Fig 8B has a somehow abnormal accumulation of CHMP5 in what looks like the midzone (it can't be intercellular bridge accumulation as it is too early in cytokinesis). Is this something representative of CHMP5 distribution during NE reformation? If not, I'd recommend to include a different image in this panel.

6) The figure legends could benefit from additional information. For example, the legend for Fig S3 needs extra labelling: there is a typo in E-D (should read E-G) and membrane deformations should be indicated with arrowheads for clarity and to highlight the results. What does time 0 represent in Fig. 8C?

Reviewer #3 (Comments to the Authors (Required)):

In this revised version, the authors have addressed most of my concerns. They have now included an analysis of the different polymeric states of Snf7 and Vps60 in yeast and imaging data of fluorescently tagged CHMP5 (Vps60) and CHMP4 (Snf7) in human cells. Unfortunately, the two tagged proteins (c-terminal or N-terminal tagged?) seem to be heavily overexpressed and their functionality has not been tested, which should make the authors very cautious when claiming they can follow the kinetics of the two proteins (Fig 7 and 8). It would be wise for the authors to acknowledge the limitations of these analysis as presented. It is also curious to see that most of the tagged CHMP5 protein seems to localize to the nucleus.

Reviewer #1 (Comments to the Authors (Required)):

The authors have submitted a marvelous revision of their original manuscript. My original concerns have been addressed. The new experiments in yeast and human cells are in line with their *in vitro* reconstitution experiments and strongly support the conclusion that Vps60 (Chmp5) and Snf7 (Chmp4) form alternative ESCRT-III filaments.

I spotted a typo in the abstract: line 22 '...may have enabled...'

We thank this reviewer for accepting our paper.

Reviewer #2 (Comments to the Authors (Required)):

In their revised article, the authors have produced new data that strengthens further the *in vitro* part of the study (new Fig4 and Sup Fig3 E-G) and show that the dynamics of assembly-disassembly for Vps60 and Snf7 are different in yeast cells (new Fig 6F-G), providing some initial cellular context to their findings. They have also investigated CHMP5 localization in mammalian cells during different ESCRT-related processes (Figs 7-8) and show that CHMP5 might not be needed for some of the pathway's cellular functions (NE reformation).

We thank this reviewer for the positive feedback on our manuscript

Comments:

1) The *in vitro* data are very strong but unequivocal demonstration of a specific function of Vps60-based filaments independently of Snf7 *in vivo* is still missing. Thus, the words "alternative" and "functionally" (distinct polymers, line 30) should be removed from the title and abstract, respectively.

We indeed agree that the data we present in this manuscript are not proving that Vps60 are functionally independent from Snf7, thus we have removed the word functionally. However, our *in vitro* data show that Vps60 can nucleate the formation of complete ESCRT-III filaments in absence of Snf7, showing that Vps60 is an alternative to Snf7 in forming ESCRT-III filaments. The role of Vps60 is similar, yet different than the one of Snf7, as some of the dynamics and properties of the Vps60-dependent filaments are different from the ones generated with Snf7. Thus we do think that the word "alternative" is still justified. It does not mean that these alternative ways of generating ESCRT-III filaments are functionally independent in the cell.

2) Can you please clarify why is Vps60 co-sedimenting at the molecular weight corresponding to the monomeric complex in the absence of functional Vps4? Wouldn't you expect to see that higher molecular weight polymers are not present (as opposed to what's observed with Snf7) but still see a polymer ~440 KD like the one in WT conditions (I'm assuming that the blots labelled as VPS4 in Fig 6F are WT cells)? How does Vps60 look in the cytoplasmic fraction?

The state of polymerization of ESCRT-III subunits is hard to assess as a whole: some subunits (chmp2/3 or vps2/vps24) can assemble in solution in absence of membranes. Snf7-chmp4 usually assemble only in presence of membranes, at least in the WT form. More importantly, how Vps4 activity affects the polymeric state again varies a lot. The general understanding of Vps4 in ESCRT-III polymeric function is that it is a depolymerization factor, but it is not entirely true. We showed that Vps4 establishes a dynamic instability, driven by ATP consumption, that participates to the growth of ESCRT-III filaments in the cell (Mierzwa et al. NCB 2017). ESCRT-III assemblies made of Snf7/Vps2/Vps24 actually grow faster with more Vps4/ATP. We also showed that this dynamic instability is the driving mechanism of the subunit sequence we observed *in vitro* and that drives membrane deformation (Pfitzner et al. Cell 2020). Thus, when removing Vps4 in the cell, it is hard to predict the final polymeric state of the subunit. The main point we wanted to make here, is that the removal of Vps4 activity, strongly affects the polymeric state of Vps60.

3) Whilst I appreciate that finding a point mutant that specifically abrogates ESCRT-III binding might not be possible, there are mutants already described that abolish the interaction between CHMP5/Vps60 and VTA1/LIP5 (doi: 10.1074/jbc.M112.417899) and finding a mutant that doesn't bind membranes should be possible based on the studies done with CHMP3 (DOI: 10.1016/j.devcel.2006.03.013). These mutants would rule out the possibility that Vps60, through its interaction with Vta1, is needed for efficient Vps4 activity on Snf7-based filaments or show that Vps60 membrane binding, independently of a possible interaction with Snf7, is needed for its function. These concerns have not been addressed.

We show in the manuscript (sup fig S4B and S5C&D) that while Vta1 has a significant effect on Vps4 activity, it is not extremely strong. While we acknowledge the importance of the previous work cited by this reviewer, and with which our results are fully consistent, we do not think that the modest effect that we see adding Vta1 in our assay justifies for a detailed study of this interaction. Also, we want to strengthen the point that Chmp3, like its yeast homolog Vs24, does not work alone, but in a subcomplex with Chmp2/Vps2. But it seems that the biochemical function of the Vps2/Vps24 vs Chmp2/3 are slightly different, in particular regarding the membrane binding, as Vps2/Vps24 does not bind membrane alone (it needs Snf7, see Mierzwa et al. NCB2017), unless incubated for

very long times, whereas the work of the Weissenhorn group has shown that Chmp2/3 binds tubular membrane from inside (Lata et al. Science 2008, and several more recent papers). So defining binding mutants of Vps24 from chmp3 may not be that straight forward. Finally, the questions asked by the reviewer about the role of Vta1 and Vps60/Snf7 interactions are certainly interesting, especially based on the previous work done on Vta1, and will be addressed in further work from our group in the future.

4) Could the fact that the dynamic of CHMP5 recruitment is delayed when compared to that of CHMP4B mean that CHMP5 is simply downstream of CHMP4B rather than segregated from CHMP4B? For example, as shown in Fig. 8C, CHMP5 is recruited to the intercellular bridge later than CHMP4B (CHMP4B appears in frame 32 min, whilst CHMP5 in 56 min) and this could similarly explain the delayed kinetics of CHMP5 recruitment to endosomes. Experiments with live cells co-expressing CHMP4B and CHMP5 (like the ones used in Fig 8 B&C) would have been better controlled in order to compare recruitment dynamics of these two proteins.

Based on our *in vitro* findings, we can say that Vps60 has the capacity to nucleate full ESCRT-III filaments on its own, without the need of Snf7. *In vivo*, several interpretations, among which the one of this reviewer, could be made to explain the delay of recruitment of CHMP5 when compared to CHMP4. Our goal in adding those data was to show that CHMP4 and CHMP5 have similar localization in the cell, while different dynamics. The exact role of CHMP5 in ESCRT-III function in cells remains to be explored in much greater details and will be done in upcoming studies from our group.

5) How many cells have the authors analyzed in the experiments shown in Fig 7 and 8? Sample size should be included in the figure legend. Also, is the CHMP4C stained cell shown in Fig 8A (bottom panel) the same as the one shown in the +LLOMe panel in Fig 7C? If so, this should be indicated in the figure legend. The cell shown in Fig 8B has a somehow abnormal accumulation of CHMP5 in what looks like the midzone (it can't be intercellular bridge accumulation as it is too early in cytokinesis). Is this something representative of CHMP5 distribution during NE reformation? If not, I'd recommend to include a different image in this panel.

We included the number of analyzed cells (ROI) and repetitions of the experiments (n=3) in the figure legends. For quantifications, 400 cells (7D) or 2500 cells (7E), respectively, were used. The zoom-in of Fig 8A shows the same cells shown in Fig 7C. We have now indicated this in the figure legend. Formation of large aggregates is a common phenotype observed upon expression of tagged ESCRT-III subunits. We find this to be true for our live-imaging experiments with CHMP5. We understand that the position of the aggregate in figure 8B might be confusing, and we thus now show the complete NE reformation and following cytokinetic abscission in a single cell (previous Fig 8C).

6) The figure legends could benefit from additional information. For example, the legend for Fig S3 needs extra labelling: there is a typo in E-D (should read E-G) and membrane deformations should be indicated with arrowheads for clarity and to highlight the results. What does time 0 represent in Fig. 8C?

We corrected the typo and extended the figure legend.

Reviewer #3 (Comments to the Authors (Required)):

In this revised version, the authors have addressed most of my concerns. They have now included an analysis of the different polymeric states of Snf7 and Vps60 in yeast and imaging data of fluorescently tagged CHMP5 (Vps60) and CHMP4 (Snf7) in human cells. Unfortunately, the two tagged proteins (c-terminal or N-terminal tagged?) seem to be heavily overexpressed and their functionality has not been tested, which should make the authors very cautious when claiming they can follow the kinetics of the two proteins (Fig 7 and 8). It would be wise for the authors to acknowledge the limitations of these analysis as presented. It is also curious to see that most of the tagged CHMP5 protein seems to localize to the nucleus.

We thank this reviewer for the support in publishing our paper, and we agree that the overexpression of the tagged proteins may alter the precise dynamics of each. We have tuned down our claims.